# The seminal odorant binding protein Obp56g is required for mating plug formation and male fertility in *Drosophila melanogaster*

**Nora C Brown[1], Benjamin Gordon[1†], Caitlin E McDonough-Goldstein[2], Snigdha Misra[1‡], Geoffrey D Findlay[1,3], Andrew G Clark[1]\*, Mariana Federica Wolfner[1]\***

[1]Department of Molecular Biology and Genetics, Cornell University, Ithaca, United States; [2]Department of Evolutionary Biology, University of Vienna, Vienna, Austria; [3]Department of Biology, College of the Holy Cross, Worcester, United States

**\*For correspondence:**
ac347@cornell.edu (AGC);
mfw5@cornell.edu (MFW)

**Present address:** †Department of Physiology and Biophysics, University of Illinois College of Medicine, Chicago, United States; ‡University of Petroleum and Energy Studies, Dehradun, India

**Competing interest:** The authors declare that no competing interests exist.

**Abstract** In *Drosophila melanogaster* and other insects, the seminal fluid proteins (SFPs) and male sex pheromones that enter the female with sperm during mating are essential for fertility and induce profound post-mating effects on female physiology. The SFPs in *D. melanogaster* and other taxa include several members of the large gene family known as odorant binding proteins (Obps). Work in *Drosophila* has shown that some *Obp* genes are highly expressed in the antennae and can mediate behavioral responses to odorants, potentially by binding and carrying these molecules to odorant receptors. These observations have led to the hypothesis that the seminal Obps might act as molecular carriers for pheromones or other compounds important for male fertility, though functional evidence in any species is lacking. Here, we used functional genetics to test the role of the seven seminal Obps in *D. melanogaster* fertility and the post-mating response (PMR). We found that *Obp56g* is required for male fertility and the induction of the PMR, whereas the other six genes are dispensable. We found males lacking *Obp56g* fail to form a mating plug in the mated female's reproductive tract, leading to ejaculate loss and reduced sperm storage, likely due to its expression in the male ejaculatory bulb. We also examined the evolutionary history of these seminal *Obp* genes, as several studies have documented rapid evolution and turnover of SFP genes across taxa. We found extensive lability in gene copy number and evidence of positive selection acting on two genes, *Obp22a* and *Obp51a*. Comparative RNAseq data from the male reproductive tract of multiple *Drosophila* species revealed that *Obp56g* shows high male reproductive tract expression in a subset of taxa, though conserved head expression across the phylogeny. Together, these functional and expression data suggest that *Obp56g* may have been co-opted for a reproductive function over evolutionary time.

## Editor's evaluation

This important study describes an atypical role of the odorant binding protein Obp56g in mating plug formation in *Drosophila melanogaster* suggesting that Obps may play roles in reproduction in addition to their originally described roles in olfaction. Mutant males lacking Obp56g fail to induce the formation of a mating plug in the female reproductive tract-leading to ejaculate loss and reduced sperm storage. The evidence supporting the claims of the authors is solid and the work will be of interest to biologists studying Obps and seminal fluid protein function and their evolution.

## Introduction

In many taxa, males transfer non-sperm seminal fluid proteins (SFPs) in the ejaculate to females during mating. Odorant binding proteins (Obps) are a common class of SFPs, and have been found in the seminal fluid (or expressed in male reproductive tissues) in a variety of invertebrate species such as mosquitoes (*Sirot et al., 2008*), honeybees (*Baer et al., 2012*), flour beetles (*Xu et al., 2013*), bollworm moths (*Sun et al., 2012*), tsetse flies (*Savini et al., 2021*), and *Drosophila* (*Begun et al., 2006*; *Findlay et al., 2008*; *Karr et al., 2019*; *Kelleher et al., 2009*). Obps have also been described in the seminal fluid of rabbits and the vaginal fluid of hamsters, although vertebrate and insect *Obp* genes are considered non-homologous and have different structures (*Mastrogiacomo et al., 2014*; *Singer et al., 1986*; *Vieira and Rozas, 2011*). Despite their widespread appearance in male seminal fluid across species, the reproductive functions of these Obps are entirely uncharacterized.

In *Drosophila melanogaster*, there are 52 members of the *Obp* gene family, many of which are highly expressed and extremely abundant in olfactory tissues such as antennae and maxillary palps (*Rihani et al., 2021*; *Sun et al., 2018*; *Vieira and Rozas, 2011*). In contrast to odorant receptors, several of which respond to specific odorants in vivo, Obps are less well characterized functionally (*Ai et al., 2010*; *Gomez-Diaz et al., 2013*; *Ha and Smith, 2006*; *Hallem and Carlson, 2006*; *Jeong et al., 2013*; *Sun et al., 2018*; *Xiao et al., 2019*; *Xu et al., 2005*). Some Obps bind odorants in vitro, and mutants of *Obp76a* (*lush*) show abnormal behavioral responses to alcohols and the male sex pheromone cis-vaccenyl acetate (cVA) (*Billeter and Levine, 2015*; *Kim et al., 1998*; *Xu et al., 2005*). These data, combined with the presence of Obps in the aqueous sensillar lymph that surrounds the dendrites of odorant receptor neurons, have led to the model that Obps bind hydrophobic odorants and help transport them across the lymph to their receptors (reviewed in *Rihani et al., 2021*). However, recent functional data demonstrating robust olfactory responses in the absence of abundant antennal Obps complicate this model and suggest Obps may have roles beyond strictly facilitating chemosensation (*Xiao et al., 2019*).

Obps are widely divergent at the amino acid level in *Drosophila*, sharing about 20% average pairwise amino acid identity gene family-wide (*Hekmat-Scafe et al., 2002*; *Vieira et al., 2007*). However, they share a conserved pattern of 6 cysteines with conserved spacing, which contribute to the formation of disulfide bonds that stabilize the alpha-helical structure (*Rihani et al., 2021*; *Vieira et al., 2007*; *Vieira and Rozas, 2011*). Evolutionarily, divergence in *Obp* gene copy number in *Drosophila* is consistent with birth-and-death models of gene family evolution, with new members arising via duplication (*Rondón et al., 2022*; *Vieira et al., 2007*; *Vieira and Rozas, 2011*). Genic and expression divergence have been reported for several Obps across *Drosophila*, leading to the hypothesis that turnover in this family may be important for the evolution of substrate preference and niche colonization (*Kopp et al., 2008*; *Matsuo, 2008*; *Matsuo et al., 2007*; *Pal et al., 2023*; *Yasukawa et al., 2010*). However, Obps in *Drosophila* and other species have wide expression patterns in larval and adult tissues (including non-chemosensory tissues), suggesting diverse roles for these proteins beyond chemosensation (reviewed in *Rihani et al., 2021*). Indeed, *Obp28a* has been implicated as a target of regulation by the gut microbiota, which stimulates larval hematopoiesis in *Drosophila* and tsetse flies (*Benoit et al., 2017*).

In *Drosophila*, two olfactory Obps have been implicated in male mating behavior: *Obp76a* (*lush*) and *Obp56h* (*Billeter and Levine, 2015*; *Shorter et al., 2016*; *Xu et al., 2005*). In males, *lush* is required for proper chemosensation of cVA in mated females through the action of *Or67d* in T1 trichoid sensilla (*Billeter and Levine, 2015*; *Kurtovic et al., 2007*; *Laughlin et al., 2008*; *Xu et al., 2005*). Knockdown of *Obp56h* in males decreases mating latency and alters pheromone profiles, including a strong reduction in the inhibitory sex pheromone 5-tricosene (5 T), indicating *Obp56h* might be involved in sex pheromone production or detection (*Shorter et al., 2016*).

In addition to the Obps that are transferred in the seminal fluid, intriguingly, several tissues in *D. melanogaster* males produce sex-specific pheromones that are transferred to females during mating. These pheromones include oenocyte-derived 7-tricosene (7-T), ejaculatory bulb-derived cVA and (3 R,11Z,19Z)–3-acteoxy-11,19-octacosadien-1-ol (CH503), and accessory gland-derived peptide prohormones (such as Sex Peptide [SP], discussed below; *Brieger and Butterworth, 1970*; *Everaerts et al., 2010*; *Guiraudie-Capraz et al., 2007*; *Scott, 1986*; *Yew et al., 2009*). These molecules have been shown to act individually (in the case of SP and CH503) or synergistically in a blend (in the case of cVA and 7-T) to decrease the attractiveness or remating rate of females with other males (reviewed in

*Billeter and Wolfner, 2018*; *Laturney and Billeter, 2016*). The coincidence of pheromones and Obps being transferred in the seminal fluid during mating has led many to hypothesize that Obps could act as molecular carriers for these molecules in mating, though direct evidence that seminal Obps impact any aspect of female post-mating behavior is lacking.

*D. melanogaster* SFPs are produced and secreted by the tissues in the male reproductive tract, including the testes, accessory glands (AGs), ejaculatory duct (ED), and ejaculatory bulb (EB; reviewed in *Wigby et al., 2020*). Many SFPs are essential for optimal fertility and the induction of the post-mating response (PMR), a collection of behavioral and physiological changes in mated females that include increased egg laying and decreased likelihood of remating (reviewed in *Avila et al., 2011*; *Wigby et al., 2020*). The induction and maintenance of this response requires the SFPs SP and the long-term response network proteins, which act in a pathway to bind SP to sperm in the female sperm storage organs (*Findlay et al., 2014*; *Ram and Wolfner, 2009*; *Singh et al., 2018*). Disrupting the presence of sperm in storage, the transfer of SP/network proteins, or the binding and release of SP from sperm leads to a loss of the persistence of the PMR and decreased fertility of the mating pair (*Findlay et al., 2014*; *Kalb et al., 1993*; *Liu and Kubli, 2003*; *Misra et al., 2022*; *Peng et al., 2005*; *Ram and Wolfner, 2009*; *Singh et al., 2018*).

A subset of the genes that encode SFPs displays interesting evolutionary patterns in many taxa, including elevated sequence divergence consistent with positive selection (or in some cases, relaxed selection), tandem gene duplication, rapid turnover between species, and gene co-option (*Ahmed-Braimah et al., 2017*; *Begun et al., 2006*; *Begun and Lindfors, 2005*; *Findlay et al., 2009*; *Findlay et al., 2008*; *Haerty et al., 2007*; *McGeary and Findlay, 2020*; *Mueller et al., 2005*; *Patlar et al., 2021*; *Sirot et al., 2014*; *Swanson et al., 2001*; *Swanson and Vacquier, 2002*). In studies of *Drosophila*, the Obps present in the seminal fluid are composed of both overlapping and distinct sets of proteins between species, mirroring a common feature of SFP evolution: conservation of functional class despite turnover of the individual genes (*Findlay et al., 2009*; *Findlay et al., 2008*; *Karr et al., 2019*; *Kelleher et al., 2009*; *Mueller et al., 2004*). This pattern is thought to be driven by sexual selection such as sperm competition and male/female intrasexual conflict, which has been hypothesized to drive molecular arms races between or within the sexes while maintaining functionality of the reproductive system (*Avila et al., 2011*; *Sirot et al., 2015*).

Here, we investigate the evolution and reproductive function of seven *D. melanogaster* seminal Obps (Obp8a, Obp22a, Obp51a, Obp56e, Obp56f, Obp56g, and Obp56i) that have been shown to be transferred to females during mating or expressed in SFP-generating tissues (*Findlay et al., 2008*; *Sepil et al., 2019*). Using a functional genetic approach, we find that six of the seminal Obps have no or a very marginal effect on the PMR in mated females. However, one Obp, *Obp56g*, is required for full male fertility and strong induction of the PMR. We further find that *Obp56g* is expressed in the male EB, loss of *Obp56g* leads to loss of the mating plug in the female reproductive tract after mating, and this loss leads to a reduction in the number of sperm stored in the mated female. Using comparative RNAseq data across *Drosophila* species, we find that *Obp56g* has conserved expression in the head, although expression in the male reproductive tract only in subset of species, suggesting potential co-option of this protein for reproductive function over evolutionary time. Finally, we investigate the molecular evolution of the seminal Obps across a phylogeny of 22 *Drosophila* species. Our results indicate duplication and pseudogenization have played an important role in the evolution of seminal Obps, as well as recurrent positive selection acting on a subset of these genes.

## Results

### *Obp56g* is required for fecundity and regulates remating rates of mated females

To test the role of the seminal Obps in the long-term PMR, we used a co-CRISPR approach to generate individual null alleles in the following genes: *Obp56f, Obp56i, Obp56e, Obp51a, Obp22a*, and *Obp8a* (*Supplementary file 4*). Additionally, we used existing mutant and RNAi lines to perturb *Obp56g* (*Jeong et al., 2013*). Collectively, we used males of these mutant and RNAi lines to measure the effect of Obp perturbation on egg laying and remating rates of their female mates. Of the seven seminal Obps, only females mated to hemizygous *Obp56g[1]/Df(2 R)* mutant males laid significantly fewer eggs and were significantly more likely to remate, indicating a loss of the PMR (*Figure 1A, B* and *Figure 2A*

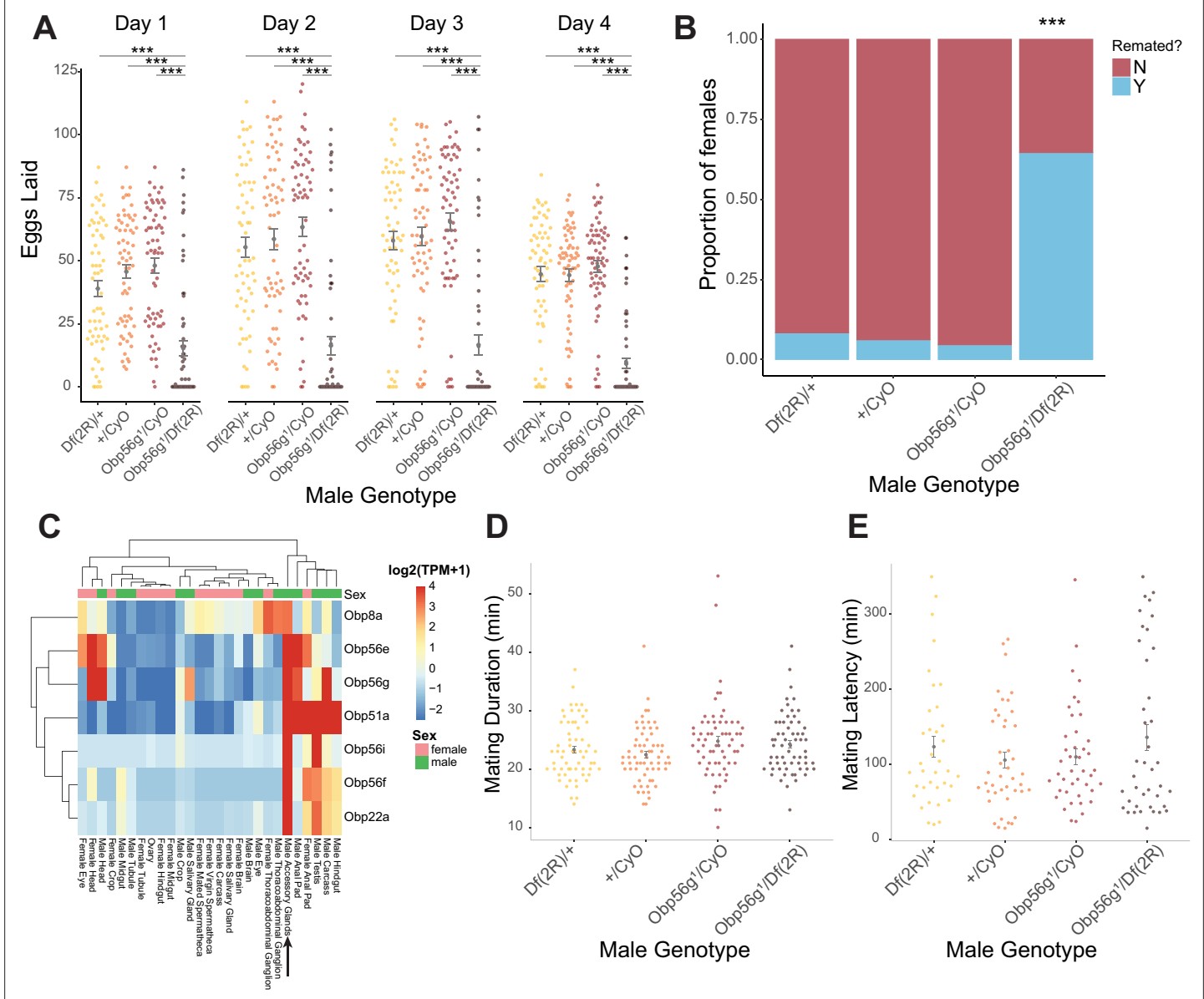

**Figure 1.** Seminal *Obp* gene expression and fecundity/remating defects in females mated to *Obp56g¹* null males. (**A**) Egg counts from CS females mated *to Df(2 R)/+, CyO/+Obp56 g¹/CyO,* or *Obp56g¹/Df(2 R)* males from 1 to 4 days after mating. Significance indicated from pairwise comparisons of male genotypes within days using emmeans on a Poisson linear mixed effects model. Error bars represent mean +/-SEM. (**B**) Proportion of females who did or did not remate with a standard CS male on the fourth day after mating within a one-hour timeframe. Significance indicated from tests of equality of proportions. (**C**) Median centered $\log_2$ normalized TPM values for the seven seminal *Obp* genes in adult tissues from FlyAtlas2.0 bulk RNAseq data. Arrow points to male accessory gland sample. (**D**) Mating duration of CS females with indicated males (all pairwise comparisons using emmeans p>0.05). (**E**) Mating latency of CS females with indicated males (all pairwise comparisons using emmeans p>0.05). For A,B, D, and E, n=61–65. Significance levels: *p<0.05, **p<0.01, ***p<0.001, n.s. not significant.

The online version of this article includes the following source data and figure supplement(s) for figure 1:

**Source data 1.** Remating counts and percentages for data shown in *Figure 1B*.

**Figure supplement 1.** Whole body knockdown of *Obp56g* using *Tubulin-GAL4* results in loss of post-mating response phenotypes in females.

**Figure supplement 1—source data 1.** Remating counts and percentages for data shown in *Figure 1—figure supplement 1B*.

**Figure supplement 2.** Male reproductive tract knockdown of *Obp56g* with *CrebA-GAL4* is required for the post-mating response and mating plug formation.

**Figure supplement 2—source data 1.** Counts and percentages for data shown in *Figure 1—figure supplement 2A and B*.

**Figure supplement 3.** Spermatogenesis appears normal in *Obp56g¹;ProtB-eGFP* males relative to *Obp56g¹/CyO;ProtB-eGFP* control males.

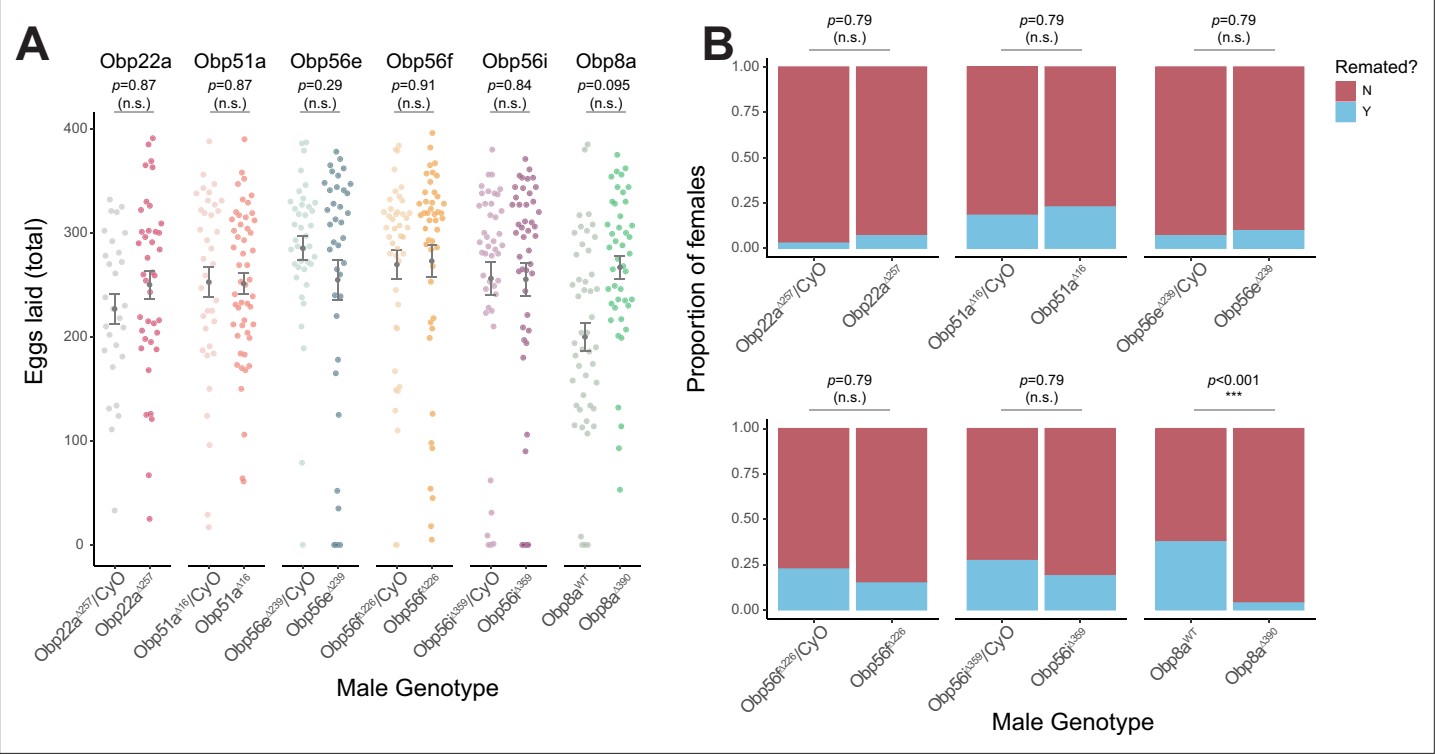

**Figure 2.** CRISPR/Cas9-generated mutants of *Obp22a, Obp51a, Obp56e, Obp56f, Obp56i,* and *Obp8a* have no or marginal effects on female fecundity and remating rates. (**A**) Egg counts from CS females mated to homozygous null or heterozygous control males (except for *Obp8a*, the control of which is from an unedited sibling line) from 1 to 4 days after mating. Significance indicated from Poisson linear models with Benjamini-Hochberg corrections for multiple comparisons. Error bars represent mean +/-SEM. (**B**) Proportion of females who did or did not remate with a standard CS male on the fourth day after mating within a one-hour timeframe. Significance indicated from Fisher's exact tests with Bejamini-Hochberg correction. Significance levels: *p<0.05, **p<0.01, ***p<0.001, n.s. not significant. For A and B, n=28–51.

The online version of this article includes the following source data and figure supplement(s) for figure 2:

**Source data 1.** Remating counts and percentages for data shown in *Figure 2B*.

**Figure supplement 1.** Crossing scheme to generate CRISPR mutants in autosomal (*Obp22a, Obp51a, Obp56e, Obp56f, Obp56i*) and X-linked (*Obp8a*) *Obp* genes used in this study, with text boxes representing chromosomes X/Y, 2, and 3 (dot chromosome not shown).

**Figure supplement 2.** No effect of heterozygosity in PMR phenotypes relative to homozygous WT or homozygous CRISPR mutant males.

**Figure supplement 2—source data 1.** Counts and percentages for data shown in *Figure 2—figure supplement 2B*.

**Figure supplement 3.** Box plots of hatchability estimates from CS females mated to *Obp56g* or CRISPR mutant males.

**Figure supplement 4.** Mating latency and duration measurements from CRISPR-generated *Obp* mutants with CS females.

**Figure supplement 5.** Mating duration (**A**) and latency (**B**) measurements from homozygous wildtype (+/+), heterozygous mutant (+/-) and homozygous CRISPR mutant (-/-) males mated to CS females.

*and B*).This phenotype was fully recessive, as heterozygous *Obp56g* mutant males (*Obp56g¹/CyO or Df(2 R)/+*) were not significantly different from *+/CyO* males, which have two copies of *Obp56g* (*Figure 1B*). We did observe slight changes in egg hatchability, although we note that the fraction of females mated to *Df(2 R)/Obp56g¹* males that laid eggs to measure hatchability from is small (*Figure 2—figure supplement 3A*). None of the other CRISPR mutant lines had a significant effect on egg hatchability, aside from a significant decrease in hatchability in the *Obp8aᵂᵀ* line (*Figure 2—figure supplement 3B*). We observed a difference in remating rates between *Obp8aᵂᵀ* and *Obp8aᐞ³⁹⁰* lines, but no difference in egg number (*Figure 2A and B*). We tested whether the autosomal CRISPR mutant males showed any effect when heterozygous by testing PMR phenotypes of wildtype (+/+), heterozygous mutant (+/-) and homozygous mutant (-/-) males, and found no statistically significant impact on egg laying or remating rate for *Obp22a, Obp51a, Obp56e, Obp56f,* or *Obp56i* (*Figure 2—figure supplement 2*).

Given that *Obp56g* is expressed in male head tissues (*Figure 1C*), we tested whether decreased mating duration could account for the decrease in fecundity in females mated to *Obp56g¹* mutant males, and found no significant difference among the four genotypes tested (*Figure 2D*. Additionally, these males do not differ in mating latency *Figure 2E*), suggesting that *Df(2 R)/Obp56g¹* males did not have baseline defects that could explain their poor induction of PMR phenotypes in females. Furthermore, sperm production in the testis of *Obp56g¹* mutant males appears normal relative to *Obp56g¹/CyO* control males, indicating the lack of fertility is not related to a spermatogenesis defect (*Figure 1—figure supplement 3*). Ubiquitous RNAi knockdown of *Obp56g* in males using a *Tubulin*-GAL4 driver recapitulated the phenotype of the hemizygous (*Obp56g¹/Df(2 R)*) mutant, resulting in decreased female egg laying and increased remating rates (*Figure 1—figure supplement 1*).

*Shorter et al., 2016* reported that male-specific knockdown of *Obp56h*, a paralogous *Obp* gene in the same genomic cluster as *Obp56e*, *Obp56f*, *Obp56g*, and *Obp56i*, shortened mating latency times; KD males were faster to mate than control males. RNAseq expression data from the FlyAtlas2.0 database shows that some of the seminal Obps are co-expressed in other tissues outside of the male reproductive tract, including head tissues (*Figure 1C*), so we tested whether our mutant lines showed altered mating latency or duration. We did not find any significant differences in either mating latency or duration in any of our mutant lines when comparing homozygous mutant males with balancer siblings, aside from a small but statistically significant decrease in mating duration in *Obp8a^{WT}* flies (*Figure 2—figure supplement 4*). Comparisons of latency and duration in (+/+), (+/-), and (-/-) CRISPR mutant males resulted in largely consistent results, with no effect on either phenotype for Obp56f, Obp22a, Obp51a, and no effect on mating duration for Obp56e (*Figure 2—figure supplement 5*). However, we did observe a slight increase in mating latency (-/- vs. +/-) and a slight decrease in duration (-/- vs. +/-and -/- vs. +/+) for Obp56i, and a slight increase in latency (-/- vs. +/+) for Obp56e (*Figure 2—figure supplement 5*).

## *Obp56g* is expressed in the *D. melanogaster* male ejaculatory bulb

While the RNAseq data shown in *Figure 1A* suggested that *Obp56g* is expressed in the male AG, *Findlay et al., 2008* reported that when females are mated to DTA-E males, which are spermless and do not produce main cell AG-derived SFPs (*Kalb et al., 1993*), transfer of all seminal Obps is lost except for Obp56g. These proteomic data suggest that Obp56g is derived from another (or an additional) tissue within the male reproductive tract. To determine where *Obp56g* is expressed in the male reproductive tract, we crossed the *Obp56g¹* mutant line (which is a promoter trap GAL4 line) to *UAS-CD4-tdGFP*. We replicated previously published expression patterns for *Obp56g* in the labellum of the proboscis (*Figure 3—figure supplement 1*), indicating that the promoter-trap GAL4 transgene should recapitulate the true expression patterns of endogenous *Obp56g* (*Galindo and Smith, 2001*). When we dissected and imaged male reproductive tracts from *Obp56g-GAL4>UAS-CD4-tdGFP* males, we observed strong GFP signal in the EB epithelium (*Figure 3A*). The EB-derived seminal protein PEB-me (also known as *Ebp*) is known to autofluoresce, resulting in autofluorescence of the tissue itself, but the GFP signal we observed in *Obp56g-GAL4>UAS-CD4-tdGFP* males is much stronger than *UAS-CD4-tdGFP* control males (*Figure 3B*; *Cohen and Wolfner, 2018*).

To determine expression patterns for the other seminal Obps, we analyzed previously published single-nucleus RNAseq data of the male reproductive tract tissues from the Fly Cell Atlas (*Li et al., 2022*). Using this approach, we confirmed that *Obp56g* is highly expressed in the EB, although we also observed expression in the ED and male AGs (*Figure 3—figure supplement 2B-D*), suggesting the promoter trap does not fully recapitulate *Obp56g* expression in all reproductive tract tissues (*Figure 3B*). For the other six Obp genes, we observed expression primarily in the AG (*Obp22a*, *Obp56e*, *Obp56i*, *Obp8a*, *Obp56f*) or ED (*Obp51a*) (*Figure 3—figure supplement 2B, C*).

## *Obp56g* is involved in mating plug formation, ejaculate retention, and sperm storage

Increased egg laying and decreased remating are two phenotypes of the PMR that depend on the presence of sperm and SP within the female sperm storage organs (*Manning, 1967*; *Peng et al., 2005*). Given that *Obp56g* is expressed in the EB, and the loss of the PMR in *Obp56g* mutant and knockdown males (*Figure 1*), we wondered whether this loss of fertility could be due to defects in mating plug formation or sperm storage. In *Drosophila*, the mating plug forms in the bursa during mating and acts

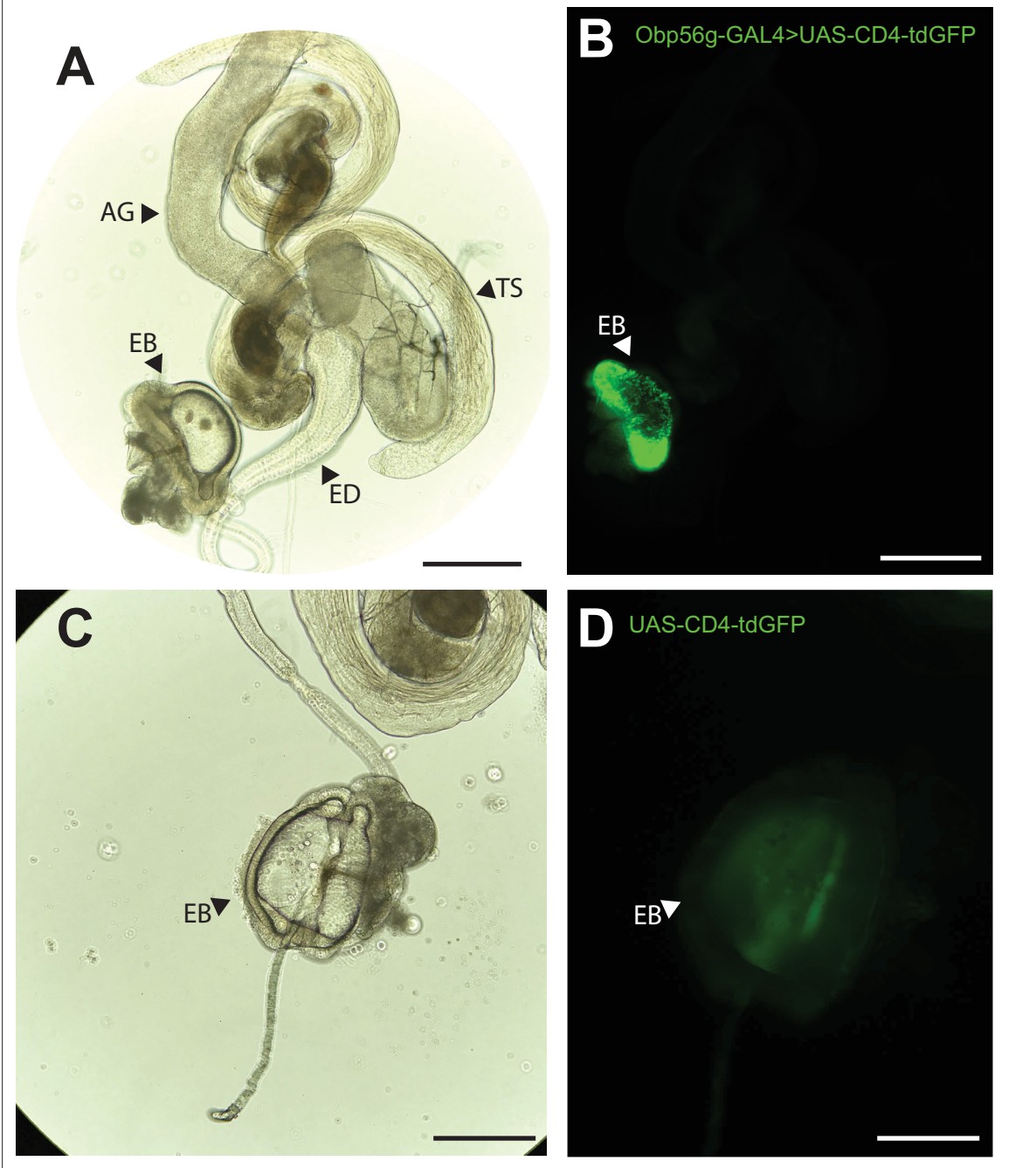

**Figure 3.** *Obp56g* is expressed in the *Drosophila* male ejaculatory bulb of the reproductive tract. (**A**) Brightfield and (**B**) GFP fluorescent microscopy image of a reproductive tract dissected from a *Obp56g-GAL4>UAS-CD4-tdGFP* male, where the following tissues are labeled: AG, accessory gland. TS, testes. ED, ejaculatory duct. EB, ejaculatory bulb. (**C**) Brightfield and (**D**) GFP fluorescent microscopy images from *UAS-CD4-tdGFP* control males, showing only the EB portion of the tract. Scale bars in A&B=130 um, C&D=70 μm.

The online version of this article includes the following figure supplement(s) for figure 3:

**Figure supplement 1.** Expression of *Obp56g-GAL4* in the gustatory bristles of the labellum.

**Figure supplement 2.** *Obp56g* is the most highly expressed seminal Obp in the ejaculatory bulb.

to retain ejaculate/sperm within the reproductive tract, until it is actively ejected by the female hours after mating (*Avila and Wolfner, 2009*). In order to test this, we crossed a *ProtamineB*-eGFP transgene (*Manier et al., 2010*), which marks the heads of sperm with GFP, into the *Obp56g[1]* mutant line, and mated homozygous null (*Obp56g[1];ProtB-eGFP*) or control (*Obp56g[1]/CyO;ProtB-eGFP*) males to

females, and directly counted sperm in the female sperm storage organs at 12 min, 3 hr, and 4 days ASM. We also used the autofluorescent nature of PEB-me to score the presence of the mating plug in the female bursa immediately after mating (*Lung and Wolfner, 2001*; *Ludwig et al., 1991*).

In contrast to *Obp56g¹/CyO; ProtB-eGFP* control males, which form a fully coagulated mating plug in the female's bursa, we observed that homozygous *Obp56g¹/Obp56g¹; ProtB-eGFP* mutant males form much less prominent and non-coagulated mating plugs (*Figure 4A and B*). While the majority of females mated to control males form a mating plug, none of the females mated to *Obp56g¹/Obp56g¹; ProtB-eGFP* males had a fully formed mating plug immediately after the end of mating (*Figure 4C*). Additionally, at this time point, a subset of females mated to *Obp56g¹/Obp56g¹; ProtB-eGFP* males lacked a sperm mass and had very few or no sperm in their bursa (*Figure 4C*). To test the possibility that *Obp56g* mutant males have defective sperm transfer, we dissected reproductive tracts from females that had been flash frozen while the flies were still copulating, 12 min ASM. In *D. melanogaster*, transfer of mating plug components, SFPs, and sperm begins at 3–5, 3, and 7 min, respectively, and is completed by 10 min ASM (*Gilchrist and Partridge, 2000*; *Lung and Wolfner, 2001*). At this time point, we noted the presence of sperm in the bursa of all females mated to both *Obp56g¹/Obp56g¹; ProtB-eGFP* and *Obp56g¹/CyO; ProtB-eGFP* males, suggesting the lack of sperm masses immediately after mating is not related to sperm transfer (*Figure 4—figure supplement 1*). Furthermore, we observed no difference in the number of sperm present in the bursa at this time point (*Figure 4D*). Rather, all females mated to *Obp56g¹/Obp56g¹; ProtB-eGFP* males lacked proper mating plugs at this time point, suggesting loss of the sperm mass is related to issues with ejaculate retention (*Figure 4—figure supplement 1*). Mutations in the other *Obp* genes had no effect on mating plug formation (*Supplementary file 5*).

Previous studies of *D. melanogaster* mating plug proteins Acp36DE and PEB-me reported a reduction in sperm storage when these genes were mutated or knocked down, indicating that integrity of the mating plug is essential for effective sperm storage (*Avila et al., 2015*; *Avila and Wolfner, 2009*; *Bertram et al., 1996*; *Neubaum and Wolfner, 1999*). At 3 hr and 4 days ASM, we observed that females mated to *Obp56g¹/Obp56g¹; ProtB-eGFP* males have significantly fewer sperm in their sperm storage organs than females mated to *Obp56g¹/CyO; ProtB-eGFP* males, (3 hr mean sperm number *Obp56g¹/CyO*: 393, mean sperm number *Obp56g¹*: 258 p<0.01; 4-day mean sperm number *Obp56g¹/CyO*: 112, mean sperm number *Obp56g¹*: 13, p<0.001 *Figure 4D*). These results suggest that the reduction in fecundity we observed in our mating assays is due to issues with sperm retention and subsequent long-term storage in *Obp56g¹* mutant males.

We further tested whether male reproductive tract expression of *Obp56g* is required for fertility and mating plug formation by knocking down *Obp56g* using a *CrebA*-GAL4 enhancer-trap driver, which drives expression in the ED and EB (*Avila et al., 2015*). We observed that mates of knockdown males showed significantly reduced egg laying and increased remating rates compared to control males, similar to whole body *Obp56g* knockdown and the *Obp56g¹* mutant line (*Figure 1—figure supplement 2A, C*). Additionally, experimental knockdown males had decreased incidence of mating plug formation compared to control males (*Figure 1—figure supplement 2B*). We also observed instances of ejaculate loss from the bursa of the female after the flies uncoupled, similar to the phenotype previously observed for *PEB-me* knockdown (*Figure 1—figure supplement 2D*; *Avila et al., 2015*). Together, these findings show that ED/EB expression of *Obp56g* is required for mating plug formation, sperm storage, and the PMR.

We next tested the possibility that Obp56g may act as a molecular carrier for seminal proteins that promote mating plug formation or the establishment of the PMR, such as SP. In order to test whether loss of *Obp56g* leads to a loss of particular SFPs in the female reproductive tract after mating, we performed western blotting on dissected female bursae samples 35 min ASM and probed for several SFPs known to be important either for the long-term PMR or mating plug formation (*Avila and Wolfner, 2009*; *Findlay et al., 2014*). We observed no difference in the synthesis of any tested protein in the male reproductive tract between *Obp56g¹/Df(2 R)* and *Obp56g¹/CyO* males (*Figure 4—figure supplement 2A, lanes 2 and 3*). Rather, we observed a lower signal intensity relative to controls in the bursa of females mated to *Obp56g¹/Df(2 R)* males for Acp36DE (and its cleavage products) at 35 min ASM, consistent with a defect in ejaculate retention in the mutant condition (*Figure 4—figure supplement 2A, lanes 4 and 5, and B*). In no case did we observe complete loss of any single protein in females mated to *Obp56g¹/Df(2 R)* males, suggesting that

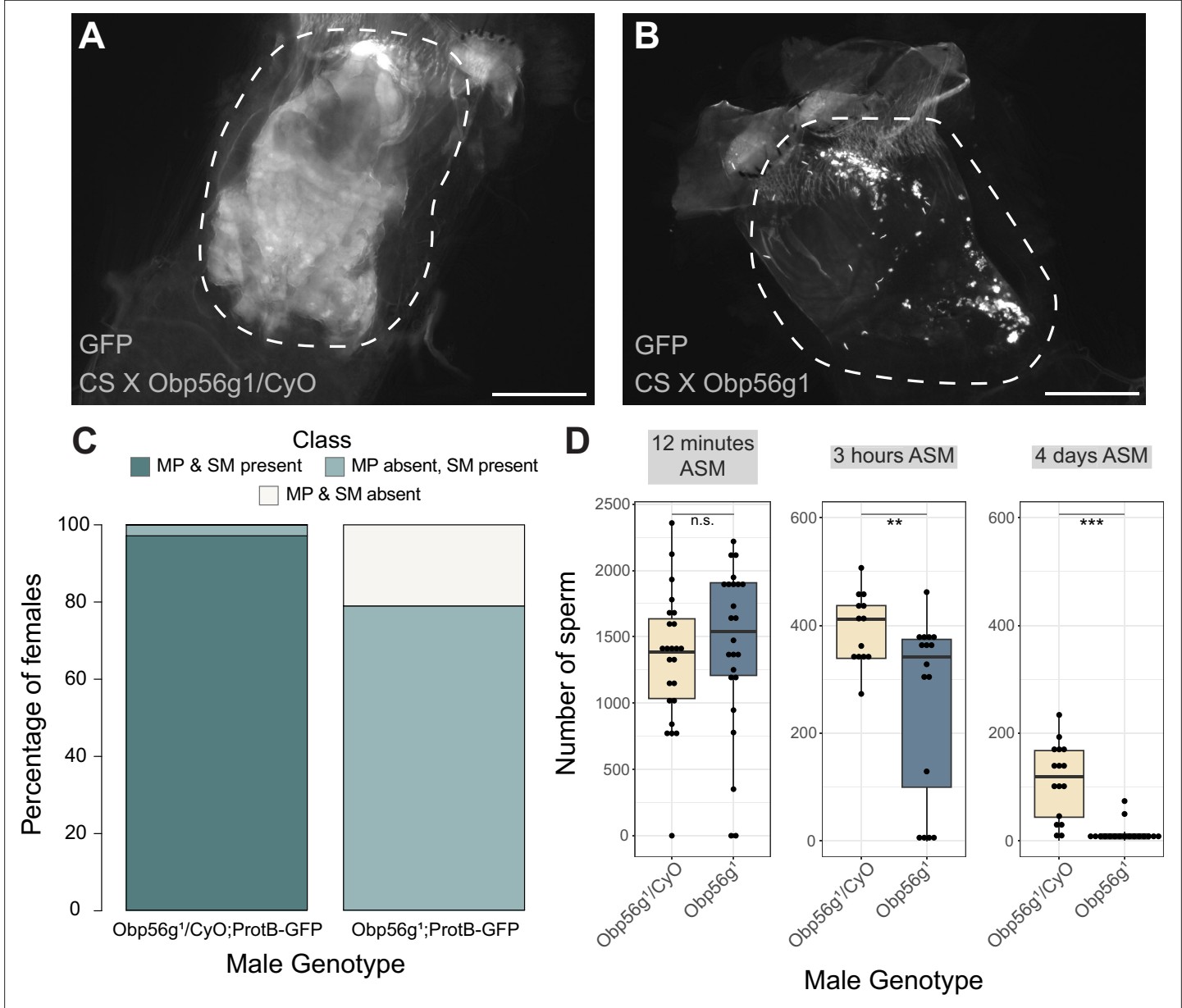

**Figure 4.** Females mated to *Obp56g1* null males have defects in mating plug formation and sperm storage after mating. (**A**) Fluorescent GFP microscopy image of the bursa of a CS female mated to a *Obp56g1/CyO;ProtB-eGFP* control male, with the mating plug surrounded by a dotted white line. Females were frozen in liquid nitrogen immediately after the end of mating. The mating plug is autofluorescent. (**B**) Fluorescent GFP microscopy image of the bursa of a CS female mated to a *Obp56g1;ProtB-eGFP* mutant male, where a similar region in the bursa as (**A**) is shown in the dotted white line. (**C**) Proportion of females mated to *Obp56g1/CyO;ProtB-eGFP* control or *Obp56g1;ProtB-eGFP* mutant males who had mating plugs or sperm masses present or absent immediately after the end of mating (n=35–38). MP, mating plug. SM, sperm mass. (**D**) Box plots of sperm counts in the storage organs of CS females mated to control (*Obp56g1/CyO;ProtB-eGFP*) or mutant (*Obp56g1;ProtB-eGFP*) males at 12 min, 3 hr, or 4 days (ASM, after the start of mating). n=13–24 for each group. Significance indicated from Student's t-tests. Significance levels: *p<0.05, **p<0.01, ***p<0.001, n.s. not significant. Scale bar = 130 µm.

The online version of this article includes the following source data and figure supplement(s) for figure 4:

**Source data 1.** Counts and proportions for data shown in *Figure 4C*.

**Figure supplement 1.** *Obp56g1* mutant males do not have gross issues with sperm transfer during mating at the 12 min ASM time point.

**Figure supplement 2.** Western blot of major SFPs from CS females mated to *Obp56g* null and control males at 35 minutes ASM.

**Figure supplement 2—source data 1.** Raw film images and uncropped, labeled western blots for data shown in *Figure 4—figure supplement 2*.

Obp56g likely does not act as the sole or an exclusive carrier for these specific proteins in the seminal fluid.

## Seminal Obps have complex evolutionary histories and exhibit evolutionary rate heterogeneity across the *Drosophila* genus

Previous studies have reported elevated rates of divergence and gene turnover of a subset of SFP genes across *Drosophila* (*Ahmed-Braimah et al., 2017*; *Begun et al., 2006*; *Begun and Lindfors, 2005*; *Findlay et al., 2008*; *Mueller et al., 2005*; *Patlar et al., 2021*; *Swanson et al., 2001*; *Wagstaff and Begun, 2005*). To examine the evolutionary history of the seminal *Obp* genes, we first identified orthologs of these genes across 22 sequenced species. Combining our orthologous gene predictions with syntenic analysis within each genome allowed us to identify several instances of lineage-specific tandem duplication and loss (*Figure 5A*, *Figure 5—figure supplements 1–6*). For example, *Obp8a* and *Obp56e* are single copy and found in most genomes across the genus, with a few predicted losses (*Figure 5A*, *Figure 5—figure supplement 2 and 4*). *Obp56f* and *Obp56i* are also single copy, though restricted to species of the *melanogaster* group (*Figure 5A*, *Figure 5—figure supplement 4 and 6*). *Obp22a* is also only found in *melanogaster* group species and has tandemly duplicated in *D. rhopaloa* and *D. takahashii* (*Figure 5—figure supplement 3*). *Obp56g* is found in all species across the genus that we examined, and has duplicated several times in the *D. willistoni* lineage to generate four copies (*Figure 5A*, *Figure 5—figure supplement 5*). Additionally, in the *obscura* group (*D. miranda, D. pseudoobscura,* and *D. persimilis*), there appears to be an intronless and highly diverged copy of *Obp56g* located immediately adjacent to the conserved gene, possibly the result of a retroduplication. *D. miranda* additionally has a putative Y-linked copy of *Obp56g* which shares 96% amino acid identity with the autosomal copy. *Obp51a*, which is only found in *melanogaster* group species, has the most extreme lability in copy number, ranging from 0 copies to 12 tandem copies in *D. eugracilis* (*Figure 5—figure supplement 1*). We also found evidence of pseudogenization events in the *Obp22a* and *Obp51a* regions in five species, which is consistent with a recent study that found evidence of pseudogenization of *Obp51a* in *repleta* group species (*Rondón et al., 2022*).

Our syntenic approach also revealed complex evolutionary events for seminal *Obp* genes not found in *D. melanogaster*. *Acp223*, a predicted Obp-like SFP gene with evidence of AG expression in *D. yakuba* and *D. erecta*, resides between *Obp56e* and *Obp56f* (*Begun et al., 2006*). InterProScan searches of this gene match signal peptide and Obp protein domains, and together with the location in the genome, suggest this gene is an *Obp56* cluster paralog (*Begun et al., 2006*). Consistent with previous reports of this gene not being present in the *D. melanogaster* genome, we were unable to find hits of this gene in *D. melanogaster* or *D. simulans* genomes using liberal E-value cutoffs in tBLASTn searches, though we found a very diverged noncoding hit in the annotated 3' UTR of *Obp56e* in *D. sechellia* (*Begun et al., 2006*). *Begun et al., 2006* reported finding a partial, noncoding orthologous region in *D. melanogaster*, which we also found in *D. simulans* to be noncoding. We did find orthologs of this gene in other *melanogaster* group species, which showed relatively long branch lengths in phylogenies of all *Obp56* cluster genes (*Figure 5—figure supplement 7A*). In the *Obp51a* cluster, we found previously reported SFPs *Sfp51D* (in *D. simulans*) and *Acp157a* (in *D. yakuba*)~14 kb upstream of *Obp51a*, which are putative orthologs of each other based on moderate branch support in our phylogenies (*Figure 5—figure supplement 7B*; *Begun et al., 2006*; *Findlay et al., 2009*). Consistent with previous results, we were unable to find orthologs of this gene in *D. melanogaster* but found a likely pseudogene in *D. simulans*. Previous work also showed this gene independently duplicated and pseudogenized in *D. yakuba* (*Begun et al., 2006*). Together, these results illustrate evolutionary lability in presence/absence and copy number of these genes in closely related *Drosophila* species.

Using our high confidence ortholog candidates, we next examined the molecular evolution of these genes across *Drosophila*. Previous reports of *Obp* gene family evolution across *Drosophila* reported heterogenous evolutionary rates for some *Obp* genes across species, but genes without 1:1 orthologs in all 12 *Drosophila* species were excluded from these previous analyses, which included *Obp51a*, *Obp22a*, *Obp56i*, and *Obp8a* (*Vieira et al., 2007*). We began by using model M0 of PAML to estimate whole-gene ratios of dN/dS ($\omega$) across all species of the phylogeny. Using this approach, we found three *Obp* genes with $\omega$ values around ~0.20 (*Obp56g*, *Obp8a*, and *Obp56e*, which are found in species beyond the *melanogaster* group, *Figure 5B*). Interestingly, the four *Obp* genes restricted

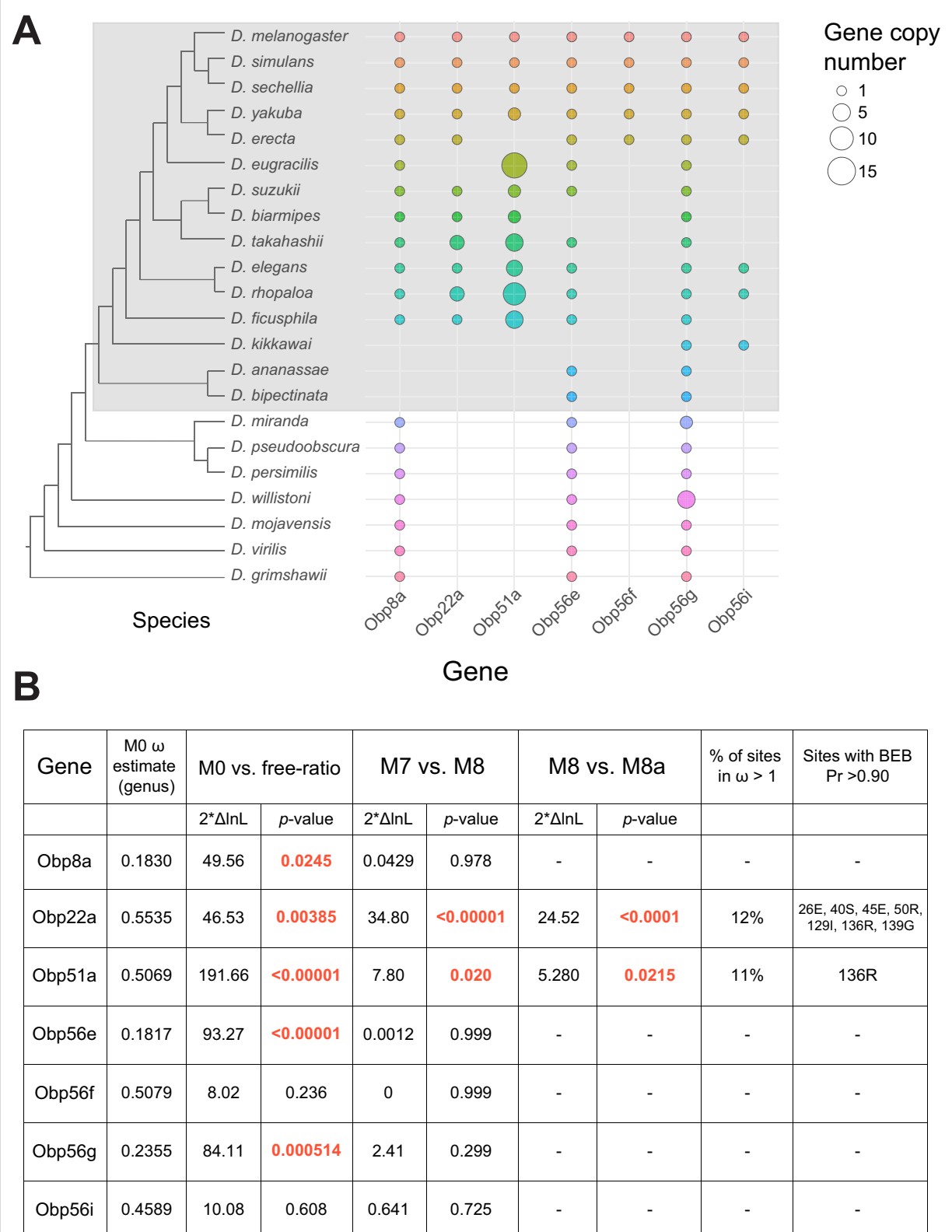

**Figure 5.** Dynamic changes in copy number, presence/absence, and evolutionary divergence rates of seminal *Obp* genes across the *Drosophila* genus. (**A**) Inferred copy number of seminal *Obp* genes across *Drosophila*. Species without a dot represent an inferred loss based on syntenic analysis. Increased size of the dot represents increased gene copy number. Phylogeny on the left from *McGeary and Findlay, 2020*. Grey box surrounds species of the *melanogaster* group. (**B**) PAML results for the seminal *Obp* genes from analysis spanning the *Drosophila* genus (M0 ω estimate, M0 vs. free ratio

*Figure 5 continued on next page*

*Figure 5 continued*

test) or spanning the *melanogaster* group (M7 vs. M8, M8 vs. M8a tests). Bold and red text indicates statistically significant comparisons. Amino acid residues with >0.90 probability of being under positive selection are indicated, with the number/letter indicative of the *D. melanogaster* position within the alignment.

The online version of this article includes the following figure supplement(s) for figure 5:

**Figure supplement 1.** Synteny plot for *Obp51a*, phylogeny on the left from **McGeary and Findlay, 2020**.

**Figure supplement 2.** Synteny plot for *Obp8a*, phylogeny on the left from **McGeary and Findlay, 2020**.

**Figure supplement 3.** Synteny plot for *Obp22a*, phylogeny on the left from **McGeary and Findlay, 2020**.

**Figure supplement 4.** Synteny plot for *Obp56e* and *Obp56f*, phylogeny on the left from **McGeary and Findlay, 2020**.

**Figure supplement 5.** Synteny plot for *Obp56g*, phylogeny on the left from **McGeary and Findlay, 2020**.

**Figure supplement 6.** Synteny plot for *Obp56i*, phylogeny on the left from **McGeary and Findlay, 2020**.

**Figure supplement 7.** RAXML-NG maximum likelihood inferred trees for genes in the (**A**) *Obp56* cluster across *melanogaster* group species, or (**B**) *Obp51a* cluster, where genes are colored as in **Figure 5—figure supplement 1**.

**Figure supplement 8.** Gene trees for *Obp22a* and *Obp51a*.

**Figure supplement 9.** Positively selected sites in *Obp22a* cluster on the outward-facing region of the protein.

to the *melanogaster* group had higher $\omega$ values, around ~0.50 (*Obp51a*, *Obp56f*, *Obp56i*, *Obp22a*, *Figure 5B*) which is much higher than the reported genome-wide average in *D. melanogaster* (*Chang and Malik, 2022*; *Drosophila* 12 *Clark et al., 2007*). We then used the 'free-ratio' model of PAML to test whether these genes exhibit evolutionary rate heterogeneity across the phylogeny. For all genes except *Obp56f* and *Obp56i*, we found significant evidence of heterogeneity in $\omega$ (*Figure 5B*), indicating these genes have experienced variable selective pressures (and/or variable strengths of selection) across the *Drosophila* genus.

## A subset of seminal Obps are evolving under recurrent positive selection

We next tested whether any seminal *Obp* genes show evidence of recurrent positive selection acting on a subset of sites by comparing models M7 and M8 in PAML, limiting our analysis to *melanogaster* group species to avoid synonymous site saturation. Using this approach, we found significant evidence of positive selection for *Obp22a* and *Obp51a*, while the other seminal *Obp* genes are evolving in a manner consistent with purifying selection (*Figure 5B*). *Obp22a* and *Obp51a* were also significant for the M8/M8a model comparison, implying positive selection rather than neutral divergence accounting for the rapid evolution of sites within these genes. Plotting the $\omega$ ratio inferred from the 'free-ratio' model onto gene trees for *Obp22a* and *Obp51a* shows multiple branches have $\omega$>1, including those with lineage-specific duplication events (*Figure 5—figure supplement 8*).

Previous work has found that pheromones derived from the male reproductive tract and transferred during mating rapidly turn over across the *Drosophila* clade, with many of these pheromones functioning as anti-aphrodisiacs in mated females (*Khallaf et al., 2021*). Given this observation, we were curious if we could infer specific sites under selection (and the 3D location of these sites within the protein) to determine whether we observe changes in the binding pocket of the protein that might be consistent with changes in ejaculate-derived ligands across species. We therefore used model M8 to infer specific sites under selection for *Obp22a* and *Obp51a* (*Figure 5B*). We included all detected copies of each gene in our selection analysis, which may have reduced our power to detect specific sites under selection for *Obp51a*, only one of which had posterior probability >0.90. For *Obp22a*, we inferred seven sites under selection (Pr >0.90), which we mapped onto the predicted AlphaFold structure of the protein (*Figure 5—figure supplement 9A*; *Jumper et al., 2021*). We found that these sites are located on the outside-facing region of the protein, away from the hydrophobic binding pocket, which has been found to bind hydrophobic ligands in other Obp proteins such as LUSH (*Figure 5—figure supplement 9B*; *Laughlin et al., 2008*).

## Male reproductive tract expression of *Obp56g* is derived in a subset of *Drosophila* species

Individual components of seminal fluid are known to turn over rapidly between species, though the larger biochemical classes these components fall into are conserved between species (*Mueller et al., 2005*; *Swanson et al., 2001*; *Wigby et al., 2020*). Beyond *D. melanogaster*, *Obp56g* has been detected as a seminal protein in *D. simulans*, *D. yakuba*, and *D. pseudoobscura*, but not in more distantly related *Drosophila* species whose SFPs have been characterized (*D. mojavensis*, *D. virilis*, and *D. montana*), despite the gene itself being conserved in these species (*Ahmed-Braimah et al., 2017*; *Garlovsky et al., 2020*; *Kelleher et al., 2009*). Considering our findings that *Obp56g* is required for male fertility in *melanogaster*, we were curious to see whether male reproductive tract expression of *D. melanogaster* seminal *Obps* was conserved across the *Drosophila* phylogeny. We therefore leveraged previously published RNAseq data from 8 different *Drosophila* species, focusing specifically on the male head and male reproductive tract samples, which include the AGs, EDs, EBs, and terminal genitalia (*Yang et al., 2018*). We observed significantly higher expression of *Obp56g* in the male reproductive tract of *D. melanogaster, simulans, yakuba, ananassae, persimilis,* and *pseudoobscura* species, and negligent or zero expression in *D. willistoni, virilis,* and *mojavensis* species (Wilcoxon rank sum test of *melanogaster/obscura* group vs. *repleta* and *virilis* group [excluding *willistoni* which has *Obp56g* duplications], $p < 0.001$), consistent with previous reports that *Obp56g* is a seminal protein in *melanogaster* and *obscura* group species (*Figure 6A*; *Findlay et al., 2008*; *Karr et al., 2019*). In head tissues, we observed high expression of *Obp56g* in all species (*Figure 6B*). We confirmed these expression patterns using semi-quantitative RT-PCR on dissected reproductive tract tissues from *melanogaster, ananassae, pseudoobscura, virilis*, and *mojavensis* males, which showed that *Obp56g* has conserved reproductive tract expression (in both the AG+ED and EB tissues) in the *melanogaster* and *obscura* groups, and conserved head expression across all species tested (*Figure 6—figure supplement 1*).

## Discussion

Obps have been identified as seminal fluid components in several insect taxa, although their functional importance in reproduction has remained unclear. We found that *Obp56g* is required for mating plug formation, sperm storage, and subsequent male fertility in *D. melanogaster*. Given that the PMR depends on sperm, SP, and the long-term response network proteins (*Findlay et al., 2014*; *Manning, 1967*; *Peng et al., 2005*), loss of ejaculate in *Obp56g* mutant males can explain the loss of long-term responses in females that we observed. Recent proteomic evidence has demonstrated that Obp56g is among the most highly abundant SFPs in the mating plug, supporting our inference that it is important for this process (*McDonough-Goldstein et al., 2022*). We further found *Obp56g* transcripts are primarily derived from the EB (although transcripts were also detected in the ED and AGs), which has previously documented functions in mating plug formation (*Avila et al., 2015*; *Bretman et al., 2010*; *Lung and Wolfner, 2001*). This EB/ED expression is required for mating plug formation and fertility. We note that *CrebA-GAL4* does not drive expression in the AG (*Avila et al., 2015*), suggesting that any residual expression in this tissue in these males is not sufficient to induce mating plug formation and the PMR.

There now is functional evidence for a growing list of mating plug and/or EB-derived SFPs, including Acp36DE, PEB-me, EbpII, and Obp56g (*Avila et al., 2015*; *Bretman et al., 2010*; *Neubaum and Wolfner, 1999*). Additionally, approaches such as gas chromatography-mass spectrometry and proteomics have characterized the male- and female-derived compounds and proteins that comprise the mating plug, and experiments dissecting the female tract at different time points after mating have elucidated the timeline of mating plug formation (*Avila et al., 2015*; *Gilchrist and Partridge, 2000*; *Laturney and Billeter, 2016*; *Lung and Wolfner, 2001*; *McDonough-Goldstein et al., 2022*). However, we still lack a detailed biochemical understanding of how the mating plug coagulates, as well as the specific mechanistic roles of the proteins highlighted above. Our finding that *Obp56g*[1] mutant males lack a mating plug at 12 min ASM suggests that this protein (and potentially its ligand, if it has one) likely functions relatively early and is required for full plug formation while the flies are still copulating. However, much remains unclear. For example, does Obp56g bind to and transport a hydrophobic reproductive tract-derived small molecule, as might be expected for an Obp? Does

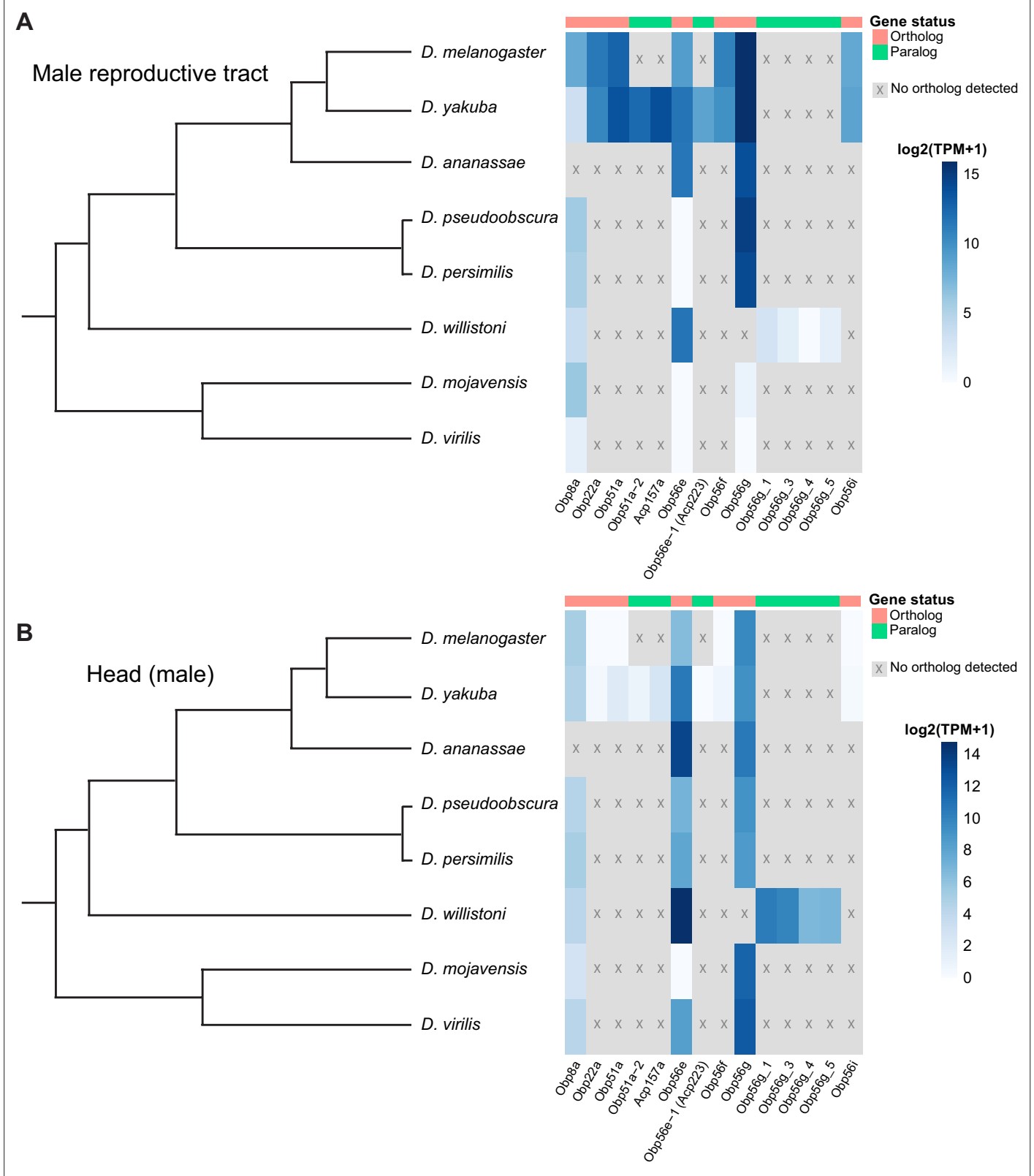

**Figure 6.** Seminal *Obp* genes show changes in expression pattern across species from bulk RNAseq data published in *Yang et al., 2018*. (**A**) log2 normalized TPM expression values (averaged across four biological replicates) of seminal *Obp* genes and their associated orthologs and paralogs in male reproductive tissue (including accessory glands, ejaculatory duct, ejaculatory bulb, and terminal genitalia for all species except *D. melanogaster*,

*Figure 6 continued on next page*

*Figure 6 continued*

which includes all tissues aside from the genitalia) of different *Drosophila* species. Grey indicates that no ortholog could be detected in that species. (**B**) log$_2$ normalized TPM expression values of seminal *Obp* gene orthologs and paralogs in male head tissue.

The online version of this article includes the following source data and figure supplement(s) for figure 6:

**Figure supplement 1.** Semi-quantitative RT-PCR data from dissected tissues (head, accessory gland +ejaculatory duct, ejaculatory bulb, and carcass) *from D. melanogaster* (Dmel), *D. ananassae* (Dana), *D. pseudoobscura* (Dpse), *D. virilis* (Dvir), and *D. mojavensis* (Dmoj) males after 35 cycles of PCR. NTC = no template control.

**Figure supplement 1—source data 1.** Raw and uncropped, labeled gel images for data shown in *Figure 6—figure supplement 1*.

Obp56g concentrate said molecule within the female tract to trigger mating plug formation, or does it merely play a structural role? Or, instead of acting as a structural component, does Obp56g signal to the female tract to secrete components that aid in mating plug formation? The answers to such questions will provide important insight into a crucial reproductive process in flies and other insect species.

*Obp56g* has interesting evolutionary characteristics in that the gene itself is conserved widely (and our results show it is under purifying selection in the *melanogaster* group), although its expression pattern in the male reproductive tract is not. Such lineage-specific shifts in expression have been reported for several other reproductive genes in *Drosophila*, including glucose dehydrogenase (*Gld*) in ED tissues of the *melanogaster* group, *jamesbond,* a fatty acid elongase responsible for CH503 production in the EB, and the Sex Peptide Receptor (*SPR*), which gained expression in the female reproductive tract in the lineage leading to the *melanogaster* group (*Cavener, 1985*; *Ng et al., 2015*; *Tsuda et al., 2015*). Our results also showed that *virilis* and *repleta* group species lack *Obp56g* expression in the male reproductive tract, which is consistent with proteomic and transcriptomic studies that did not detect *Obp56g* as a predicted seminal protein in these species (*Ahmed-Braimah et al., 2017*; *Kelleher et al., 2009*). Previous studies have described insemination reactions (*repleta* group) and 'dense copulatory plugs' (*virilis* group) in the bursa of females of these species post-mating (*Markow and Ankney, 1988*; *Patterson, 1946*). While these structures are very likely composed of ejaculate matter (and female-derived components), whether they are true homologous structures to the *melanogaster* mating plug, which has documented functional roles in promoting sperm storage and in post-mating pheromonal mate guarding, is unclear (*Avila et al., 2015*; *Avila and Wolfner, 2009*; *Laturney and Billeter, 2016*; *Neubaum and Wolfner, 1999*). A previous study using electron microscopy to analyze post-mating structures in the female bursa in *D. melanogaster* and *D. mojavensis* found the composition, density, and size of these structures to be quite distinct, and characterized them as separate phenomena (termed a 'sperm sac' and 'true insemination reaction' for *melanogaster* and *mojavensis*, respectively; *Alonso-Pimentel et al., 1994*). Interestingly, however, several recent studies have shown rapid divergence and anti-aphrodisiac function of pheromonal compounds produced in the EB or male reproductive tract across *Drosophila* (*Chin et al., 2014*; *Khallaf et al., 2021*; *Ng et al., 2014*). Elucidating the mechanistic function of *Obp56g* will provide interesting insight into whether the rapid turnover of male-specific pheromones is linked to the evolutionary changes in expression we observe for *Obp56g* and the evolutionary turnover in seminal Obps seen across more distant taxa. A further question remains whether *Obp56g* has a conserved function in mating plug formation in the species where the gene is an SFP (and its function in those where it is not), which could help elucidate when and how *Obp56g* acquired its role in reproduction. Furthermore, whether Obp56g took over a primary role in mating plug formation after it evolved reproductive tract expression, and whether "plugs" or other post-mating structures were fundamentally different prior to this, remains an open question.

Our results also show that when seminal *Obp* genes are individually knocked out, only *Obp56g* has a strong effect on the PMR and male fertility, while loss of the others has no effect (for *Obp8a*, the mutant had slightly lower remating rates than the control, which is opposite of what is expected for genes involved in PMR phenotypes—this can potentially be explained by our finding that *Obp8a*$^{WT}$ flies mated for less time than *Obp8a*$^{\Delta390}$ flies). These results can be explained in part given our findings that *Obp56g* is the only seminal Obp that is highly expressed in the EB, which has documented functions in mating plug formation. The other Obps are derived from the AG (*Obp51a, Obp22a, Obp56e, Obp56i, Obp8a*) or the ED (*Obp51a*), which is consistent with previous transcriptomic and proteomic studies of the reproductive tract (*Findlay et al., 2008*; *Li et al., 2022*; *Majane et al., 2022*; *Takemori and Yamamoto, 2009*). Alternatively, given these genes are in the same gene family,

redundancy might mask any individual gene's phenotype, and defects in fertility may only be apparent when these genes are mutated in combination. Indeed, previous studies in *Drosophila* have shown functional redundancy among paralogs of the *Obp50* cluster in male starvation resistance (*Johnstun et al., 2021*). Evolutionarily, it has been hypothesized that sexual conflict between males and females can drive functional redundancy in the biochemical classes present in seminal fluid through mechanisms of gene duplication, co-option, and gene loss, although this has never been directly functionally tested (*Sirot et al., 2015*). Alternatively, it is possible that the genes for which we did not detect a PMR phenotype are involved in another aspect of reproduction or mating. Given that we detected positive selection acting on *Obp22a* and *Obp51a*, it would be informative to test whether these genes might be involved in mediating outcomes of sperm competition, as has been observed for other SFPs that show signatures of selection (*Avila and Wolfner, 2009*; *Patlar and Civetta, 2022*; *Wong et al., 2008*). Measuring short-term remating rates (0–5 hr, before the long-term SP response becomes active) would also be informative and might be consistent with a male-derived pheromonal function for these genes (*Bretman et al., 2010*; *Laturney and Billeter, 2016*).

Given several previous studies demonstrating elevated divergence of SFP genes in *Drosophila*, we tested whether any of the seminal *Obp* genes are rapidly evolving in the *melanogaster* group. We did not detect positive selection on *Obp56g*, *Obp56e*, *Obp56f*, *Obp56i*, or *Obp8a*, but did detect positive selection acting on *Obp22a* and *Obp51a*. We found that *Obp56g* is highly expressed in head tissues across all the species we tested, raising the possibility that the gene is under pleiotropic constraint for a non-reproductive function, thus limiting its capacity to rapidly diverge (though we did observe a highly diverged paralog of *Obp56g* in the *obscura* clade). Previous studies in *D. melanogaster* have shown *Obp56g* is highly expressed in gustatory sensilla in the labellum in males and females, although functional studies of *Obp56g[1]* mutants showed they had normal attractive and aversive behaviors to sucrose and bitter-tasting compounds, respectively (*Galindo and Smith, 2001*; *Jeong et al., 2013*). In our assays, *Obp56g[1]* mutants did not have significantly altered mating latency or duration times from controls, indicating it does not play a role in male courtship behavior as measured in our assays. Thus, the proboscis-related function of *Obp56g*, and whether it is conserved across species (which would possibly explain our observations of purifying selection acting on the gene), remains unknown. Alternatively, *Obp56g* could possibly be conserved within the *melanogaster* group due to its role in mating plug formation, as it is essential for full male fertility in *D. melanogaster*. Such a hypothesis is consistent with previous findings of conservation among some members of the SP network, whose functions are necessary for successful reproduction in *melanogaster* (*McGeary and Findlay, 2020*).

Our study also revealed extensive evolutionary lability in copy number of the seminal Obps across species, which appears to be driven by tandem gene duplication, pseudogenization, and gene loss, particularly in the *Obp51a* cluster. Gene duplication has been shown to be a major force in the evolution of female reproductive tract and SFP genes, although the reasons why are less clear (*Findlay et al., 2008*). There may be selection acting on increased protein abundance, which could be accomplished by gene duplication (*Kondrashov et al., 2002*). Alternatively, models of sexual conflict propose arms race-style antagonism between males and females, whereby duplication and divergence of reproductive molecules may allow either sex to counter-adapt against the other (*Findlay et al., 2008*; *Kelleher and Markow, 2009*; *Kelleher and Pennington, 2009*; *Sirot et al., 2014*; *Swanson and Vacquier, 2002*). Our finding of positive selection acting on *Obp22a* and *Obp51a* suggests the latter may be involved. Studies have also previously demonstrated that relaxed constraint following gene duplication can allow for deleterious or complete loss of function mutations, resulting in gene loss or the formation of pseudogenes, which could explain the patterns of duplication and pseudogenization we observed in the *Obp51a* and *Obp22a* clusters (*Birchler and Yang, 2022*; *Ohno, 1970*; *Sirot et al., 2015*).

Overall, our study provides new evidence for a novel reproductive role for Obps, highlighting the broad functional diversity for this gene family in *Drosophila*. Additionally, we observed expression shifts, duplication, and divergence in the evolution of these seminal protein genes, highlighting the myriad mechanisms by which reproductive genes can diverge across species. The frequent occurrence of Obps in the seminal fluid across distinct taxa raises the possibility that members of this gene family are repeatedly co-opted into the SFP suite by various means. Functional studies of seminal Obps across these diverged species will provide important comparative data for whether seminal Obps can evolve roles in reproductive processes beyond mating plug formation.

## Materials and methods

### Fly stocks and husbandry

Flies were reared and mating assays performed on a 12 hr light/dark cycle on standard yeast/glucose media in a 25 °C temperature-controlled incubator.

We used the following lines in this study: BL#55079 (*w[\*]; TI{w[+mW.hs]=GAL4}Obp56g[1]*) (*Jeong et al., 2013*); *UAS-CD4-tdGFP* (*Han et al., 2011*); *LHm pBac{Ubnls-EGFP, ProtB-eGFP}(3)* (a gift from J. Belote and S. Pitnick, Syracuse University) (*Manier et al., 2010*); Canton-S (CS); *w1118*; BL#25678 (*w[1118]; Df(2 R)BSC594/CyO*) (*Cook et al., 2012*); *w;;Gla/CyO; w;;TM3/TM6b*; BL#3704 (*w[1118]/ Dp(1;Y)y[+]; CyO/Bl[1]; TM2/TM6B, Tb[1]*); *y1 w1118; attP2{nos-Cas9}/TM6C,Sb Tb* (*Kondo and Ueda, 2013*); BL#51324 (*w[1118]; PBac{y[+mDint2] GFP[E.3xP3]=vas-Cas9}VK00027*); VDRC#23206 (*UAS-Obp56g*RNAi from the GD library); BL#49409 (*w[1118]; P{y[+t7.7] w[+mC]=GMR64E07-GAL4} attP2*) (*Jenett et al., 2012*); *C(1)DX, y[1] w[1] f[1]/FM7c, Kr-GAL4[DC1], UAS-GFP[DC5], sn[+];;;* (a gift from Susan Younger, University of California San Francisco); *Tubulin*-GAL4 (*Findlay et al., 2014*); BL#35569 (*y[1] w[\*] P{y[+t7.7]=nos-phiC31int.NLS}X; PBac{y[+]-attP-9A}VK00027*). We obtained lines of *D. ananassae, D. pseudoobscura, D. mojavensis,* and *D. virilis* from the *Drosophila* Species Stock Center at Cornell University.

To generate males varying in numbers of copies of *Obp56g*, we used a line carrying the *Obp56g*[1] mutant allele, which is a complete replacement of the *Obp56g* coding sequence with a GAL4 mini-*white* cassette (*Jeong et al., 2013*). We crossed homozygous *Obp56g*[1] flies with *Df(2 R)BSC594/CyO* to generate trans-heterozygous *Obp56g*[1] over a deficiency of chromosome 2 R, or *Obp56g*[1] balanced over *CyO* (which have zero and one copy of functional *Obp56g*, respectively). We then crossed *w1118* (the genetic background of the *Obp56g*[1] null line) with *Df(2 R)BSC594/CyO* to obtain *+/Df(2 R)* or *+/ CyO* males (which have one and two copies of functional *Obp56g*, respectively).

To knock down expression of *Obp56g* in males, we drove a UAS-dsRNA construct against *Obp56g* (VDRC#23206) using the ubiquitous *Tubulin*-GAL4 driver (*Lee and Luo, 1999*). Control males were the progeny of UAS-*Obp56g*RNAi crossed to *w1118*.

To knock down expression of *Obp56g* in the male ED and EB, we drove UAS-*Obp56g*RNAi with a *CrebA*-GAL4 enhancer trap driver (*Avila et al., 2015*; *Jenett et al., 2012*). Control males were the progeny of *CrebA*-GAL4 crossed to *w1118*.

### Construction of gRNA-expressing lines and CRISPR genome editing

To generate individual *Obp* null alleles, we used a co-CRISPR approach to target each *Obp* gene along with the gene *ebony* as previously described for *Drosophila* (*Kane et al., 2017*). To this end, we opted for a strategy in which transgenic multiplexed gRNA expressing lines were crossed to germline Cas9 expressing lines (see *Figure 2—figure supplement 1* for full crossing scheme).

To generate our gRNA constructs, we used flyCRISPR's Optimal Target Finder tool to design three gRNAs per *Obp* gene (two guides targeting the 5' CDS of the gene, the third guide targeting the 3' end, *Supplementary file 1*; *Gratz et al., 2014*). We then integrated these gRNA sequences (and a gRNA targeting *ebony*) into pAC-U63-tgRNA-Rev, a plasmid that expresses multiplexed gRNAs under the control of the U6:3 promoter (*Supplementary files 2 and 3*, supplemental methods; *Kane et al., 2017*; *Poe et al., 2019*). The resulting plasmids were injected into BL#35569 (*y[1] w[\*] P{y[+t7.7]=nos-phiC31int.NLS}X; PBac{y[+]-attP-9A}VK00027*) embryos by Rainbow Transgenic Flies, and integrated into the third chromosome attP[VK27] site via PhiC31-mediated integration.

For the autosomal *Obp* SFP genes, each stable transgenic gRNA line was crossed to *yw;;nos-Cas9attP2* flies in the P0 generation, and the resulting P1 progeny were crossed to *w; CyO/Bl; TM2,e/ TM6B,e* as in *Kane et al., 2017*. Resulting F1 *ebony/TM6B,e* or *ebony/TM2,e* flies were backcrossed for two generations to *w;Gla/CyO* to isolate mutant *Obp* alleles (and to remove third chromosome *ebony* mutations). The *Obp* mutant lines were then maintained as a heterozygous stock over *CyO* in a *white*⁻ background (see *Figure 2—figure supplement 1* for the detailed crossing scheme). All mutations were validated using PCR and Sanger sequencing with primers that target ~150 bp upstream and downstream of each *Obp* gene *Supplementary files 3 and 4*.

For *Obp8a*, which is X-linked, the crossing scheme was the same as above except that we used *w;;vasa-Cas9* to avoid introducing *Obp* mutations on a *yellow*⁻ chromosome (*Figure 2—figure supplement 1*). Additionally, we used an *FM7c* balancer line instead of *w;Gla/CyO*.

For the mating assays, we used homozygous null *Obp* mutants (*Obp^mut^*) and their heterozygous *Obp^mut^/CyO* siblings as controls. We additionally isolated an unedited sibling line and crossed each Obp mutant line to compare homozygous wildtype, heterozygous mutant, and homozygous mutant males without the balancer chromosome. For *Obp8a* mutants, we used unedited males from sibling lines as controls.

## Verifying levels of knockdown

We used RT-PCR to assess the level of expression of *Obp56g* in our experimental and control knockdown flies. We extracted RNA from whole flies using RNAzol, treated the samples with DNase (Promega), and synthesized cDNA as previously described (*Chen et al., 2019*), (Sigma-Aldrich). *Obp56g* was then amplified via RT-PCR, using *Rpl32* as a positive control, and $dH_2O$ as a negative control. For *Obp56g* RNAi, we removed the heads of the flies prior to extracting RNA from the rest of the body, which was necessary to increase sensitivity to detect reproductive tract expression, since *Obp56g* is expressed in the head (*Galindo and Smith, 2001*; *Jeong et al., 2013*).

## Mating assays

We collected unmated flies under $CO_2$ anesthesia and aged males and females in separate vials for 3–5 days post-eclosion. We randomly assigned females to a given male genotype and observed single pair copulations, after which we removed the male using an aspirator. The experimenter was then blinded from the genotype of the male for the duration of the experiment. We discarded any mating pair that copulated for an unusually short duration (<10 min) as previously described (*LaFlamme et al., 2012*). Each mating assay was performed two to three independent times.

Mating latency was measured as the time difference between introducing the male into the vial and the beginning of mating. Mating duration was measured as the time difference between the end of mating and the beginning of mating. Time data were converted to minutes using the R package chron (version 2.3–58), and statistical differences between male genotypes were tested using Student's T-tests or linear mixed effect models in R (*James and Hornik, 2022*).

Mating assays (female egg laying, egg hatchability, and female remating rate) were performed as previously described (*Findlay et al., 2014*). For measuring remating rate, CS females were mated in single pairs to males of a given genotype, after which the male was removed. Four days later, a single CS male was added to the vial and remating was scored within a one-hour time frame. The four-day post-mating timepoint was chosen (for remating and egg counts) as it is within the window of the normal SP-mediated long-term PMR response (*Findlay et al., 2014*).

We assessed statistical significance for egg counts using a generalized linear mixed effects model using the lme4 package (version 1.1–30) in R version 4.2.1, where male genotype and day were included as fixed effects, and vial and replicate were included as a random effects, as previously described (*Bates et al., 2015*; *Findlay et al., 2014*; *LaFlamme et al., 2012*). Egg laying was modeled using a Poisson distribution, and the fit of the full model was compared against a reduced model where male genotype was dropped, using the R function aov. We accounted for false discovery rate by applying a Benjamini-Hochberg correction (*Benjamini and Hochberg, 1995*). To assess on which day differences among genotypes were significant, we performed pairwise comparisons on estimated marginal means between days and genotypes using the R package emmeans (version 1.8.1–1) (*Lenth et al., 2022*). Significance in egg hatchability was assessed the same way, except we used a binomial distribution as previously described (*LaFlamme et al., 2012*). We assessed statistical significance for differences in female remating rates between two male genotypes using Fisher's exact tests, and tests for equality of proportions when comparing across more than two male genotypes.

To assess mating plug formation and sperm storage, we crossed a *ProtamineB-eGFP* transgene (*Manier et al., 2010*) into the *Obp56g^1^* background to visualize sperm directly. We observed single pair matings between CS females and either *Obp56g^1^/Obp56g^1^; ProtB-eGFP* or *Obp56g^1^/CyO; ProtB-eGFP* males. Females were flash frozen in liquid nitrogen immediately after the end of mating. We dissected the lower female reproductive tract (including the bursa, seminal receptacle, and spermathecae) into ice cold PBS, mounted the tissue in a drop of PBS, and added a coverslip. The tissue was imaged on an ECHO-Revolve microscope using a 10 X objective with a FITC LED light cube to visualize the autofluorescent mating plug, and each female was scored as having a mating plug present or absent. Statistical significance in mating plug presence vs. absence was assessed using Fisher's exact

tests. Sperm counts using these male genotypes were performed similarly, with mated CS females flash frozen either 3 hr or 4 days after the start of mating (ASM). To facilitate sperm counting, the SR was unwound using forceps, and the spermathecal caps were gently crushed under the coverslip to release sperm. Sperm from both spermathecal caps was counted per individual. Statistical significance in sperm counts was assessed using Student's T-tests in R.

To assess sperm transfer during mating, we flash froze copulating pairs of CS females and either *Obp56g¹/Obp56g¹; ProtB-eGFP* or *Obp56g¹/CyO; ProtB-eGFP* males in liquid nitrogen 12 minutes ASM, a time point when efficient transfer of both sperm and seminal fluid components has finished (*Gilchrist and Partridge, 2000*; *Lung and Wolfner, 2001*). Frozen males and females were gently separated at the genitalia, and the female reproductive tract was dissected and scored as described above for the presence/absence of the sperm mass and mating plug, as well as sperm number.

## Expression patterns

To determine male expression patterns of *Obp56g* in the reproductive tract, we crossed the deletion line of *Obp56g* (BL#55079), which is a promoter-trap GAL4 line, to a *UAS-CD4-tdGFP* line to generate *Obp56g-GAL4>UAS-CD4-tdGFP* flies (*Jeong et al., 2013*). Unmated males were aged 3–5 days, and entire reproductive tracts were dissected into ice cold PBS. The tissue was mounted in PBS and a coverslip was added. The tissue was imaged using an ECHO-Revolve microscope as described above, using the FITC light cube to visualize live GFP fluorescence. The EB is known to autofluoresce due to the seminal protein PEB-me (*Lung and Wolfner, 2001*), so as a negative control we imaged reproductive tracts from *UAS-CD4-tdGFP* males.

We tested for expression of the other seminal Obps in different parts of the male reproductive tract using previously published single nucleus RNAseq data from the Fly Cell Atlas (*Li et al., 2022*). We used scripts from (*Raz et al., 2022*) to load the loom file, scale, and normalize the expression data from the stringent 10 X male reproductive gland sample using Seurat (version 4.2.0), SeuratDisk (version 0.0.0.9020), and ScopeLoomR (version 0.13.0) in R (*Hoffman, 2022*; *Li et al., 2022*; *Satija et al., 2015*). Differences in seminal *Obp* expression level within the EB cluster were tested using Wilcoxon rank sum tests in R.

To examine *Obp* expression patterns across species, we used publicly available RNAseq data from dissected tissues and whole bodies for the following species of *Drosophila: melanogaster, yakuba, ananassae, pseudoobscura, persimilis, willistoni, virilis,* and *mojavensis* (*Yang et al., 2018*). Gene level read counts were obtained from this study (GSE99574) based on HiSAT2 alignments to the FlyBase 2017_03 annotation. Counts were then normalized within species for genes with at least one read across all samples in DEseq2 with a median ratio method, then log2 normalized with an added count of 1.

To verify the expression patterns seen in the RNAseq dataset, and to determine which tissue of the reproductive tract was responsible for expression, we performed semi-quantitative RT-PCR for *Obp56g* from dissected heads, AGs, EBs, and carcasses from males of *Drosophila* species: *melanogaster, ananassae, pseudoobscura, virilis,* and *mojavensis*. For each species, we reared flies and separated males and females under $CO_2$ anesthesia and aged the males to sexual maturity (*Ahmed-Braimah et al., 2017*; *Karr et al., 2019*; *Kelleher et al., 2009*; *Tsuda et al., 2015*). We dissected tissues from ~25 males directly into RNAzol, and prepared cDNA as described above. We designed species-specific primers for *Obp56g* (*Supplementary file 3*) and used *Actin5C* and $dH_2O$ controls.

## Western blotting

To assess the production and transfer of specific seminal proteins, we performed Western blotting on protein extracts from CS females that were mated to either experimental *Obp56g¹/Df(2 R)* or control *Obp56g¹/CyO* males and flash frozen in liquid nitrogen 35 min ASM. For each genotype, we dissected the reproductive tracts from 1 male and 4 mated CS females and performed Western blotting using antibodies against SP, CG1656, CG1652, Antares (Antr), CG9997, CG17575, Acp36DE, Ovulin (Acp26Aa), and tubulin as a loading control as previously described (*Misra and Wolfner, 2020*). Protein extracts were separated on a 12% acrylamide gel, transferred to PVDF membranes, and probed for each seminal protein. Antibodies were used at the following concentrations: Acp26Aa (1:5000), Acp36DE (1:12,000), Antr (1:750), CG9997 (1:750), SP (1:1,000), CG1652 (1:250), CG1656 (1:500), CG17575 (1:500), Tubulin (1:4000, Sigma-Aldrich T5168) (*LaFlamme et al., 2012*; *Ram and*

*Wolfner, 2009*; *Singh et al., 2018*). Band intensity was measured in Image Studio Lite (LI-COR Biosciences) and normalized to the tubulin band within a sample. Statistical significance of male genotype was assessed using a linear model as described above.

## Evolutionary analysis

We obtained orthologous coding sequences for each of the seminal Obps from the following 22 *Drosophila* species from NCBI: *melanogaster, simulans, sechellia, erecta, yakuba, ananassae, eugracilis, suzukii, biarmipies, takahashii, elegans, rhopaloa, ficusphila, kikawaii, bipectinata, miranda, pseudoobscura, persimilis, virilis, willistoni, mojavensis,* and *grimshawi*. To do so, we used gene ortholog predictions from the *Drosophila* evolutionary rate covariation ortholog dataset, which was generated using the OrthoFinder2 algorithm (*Findlay et al., 2014*; *Raza et al., 2019*). To bolster our ortholog predictions, we performed reciprocal best tBLASTn searches in each of the genomes using the focal *D. melanogaster Obp* gene as the query, retaining only those genes that were reciprocal best hits for study (this filtered ~24% of the predicted orthologs, which were frequently evolutionarily older paralogs from the same genomic cluster). For orthologous gene groups with predicted paralogs, we identified the syntenic region in the target genome by finding orthologs of the flanking genes, assuming conservation of gene order. Additionally, we used RAxML-NG to construct maximum-likelihood phylogenies from the predicted coding sequences to further validate orthology calls for genes with predicted paralogs (*Kozlov et al., 2019*). Using this syntenic approach, we identified instances where some genes were unannotated by the NCBI Gnomon pipeline. In these situations, we ensured the unannotated genes we retained for our evolutionary analysis had intact open reading frames, splice sites, and lacked premature stop codons. We additionally used InterProScan to ensure these genes had a predicted Obp protein domain (*Jones et al., 2014*).

We used MUSCLE implemented in MEGA-11 with default settings to align the amino acid sequences, and back-translated the alignment obtain the cDNA alignment (*Edgar, 2004*; *Tamura et al., 2021*). We constructed a consensus phylogeny based on a concatenated nucleotide alignment of the *Obp* genes using RAxML-NG, where gaps were used when a particular protein was missing from a species as previously described (*Kozlov et al., 2019*; *McGeary and Findlay, 2020*). *Obp51a* was excluded from this concatenated tree due to extensive tandem gene duplication. In RAxML-NG, we used the GTR + Gamma models and performed non-parametric bootstrapping with 1,000 replicates (*Kozlov et al., 2019*). We used the Transfer Bootstrap Expectation (TBE) as a branch support metric as previously described (*Carlisle et al., 2022*). We used the top scoring tree topology from RAxML-NG for all analyses run in PAML for genes predicted to be single copy across the *melanogaster* group. For genes with duplications in the *melanogaster* group (*Obp22a* and *Obp51a*), we also constructed gene trees using RAxML-NG, and used those phylogenies in PAML.

For our evolutionary analyses, we used the codeml package in PAML to run branch and sites tests (*Edgar, 2004*; *Kumar et al., 2018*; *Yang, 2007*). For the branch test, we used the consensus phylogeny for all 22 species and compared the likelihood ratio of the 'free ratio' model with the M0 model. For the sites tests, we limited species in the analysis to those in the *melanogaster* group to avoid saturation of synonymous sites. For these analyses, we used likelihood ratio tests to compare the M7 with the M8 model. For those genes which showed evidence of positive selection in the M7 vs. M8 comparison, we then performed likelihood ratio tests between models M8 and M8a. For genes in which the M8 model was a significantly better fit, we then used the Bayes empirical Bayes (BEB) predictions to identify specific sites under positive selection. For any genes with significant evidence of positive selection, we detected recombination breakpoints in the *Obp* genes using GARD implemented in DataMonkey, partitioned the genes at the breakpoints and re-ran PAML on each segment separately as previously described (*Kosakovsky Pond et al., 2006*; *McGeary and Findlay, 2020*).

## Materials availability statement

All new CRISPR mutants and gRNA lines generated for this study are available upon request.

## Acknowledgements

We thank Dr. Yasir Ahmed-Braimah for help analyzing FlyAtlas2.0 data, Dr. Jolie Carlisle for help with the evolutionary analysis, Norene Buehner for help with Western blots, and members of the Wolfner and Clark labs for useful comments and advice. We also thank Susan Younger, J Belote and

S Pitnick, the Vienna *Drosophila* Resource Center, the Bloomington *Drosophila* Stock Center, and the *Drosophila* Species Stock Center for lines. This work was supported by NIH grant R01-HD059060 to AGC and MFW, NIH postdoctoral fellowship F32GM097789 to GDF, and NSF grant 2212972 to GDF.

# Additional information

## Funding

| Funder | Grant reference number | Author |
|---|---|---|
| National Institutes of Health | HD059060 | Andrew G Clark<br>Mariana Federica Wolfner |
| National Institutes of Health | F32GM097789 | Geoffrey D Findlay |
| National Science Foundation | 2212972 | Geoffrey D Findlay |

The funders had no role in study design, data collection and interpretation, or the decision to submit the work for publication.

## Author contributions

Nora C Brown, Benjamin Gordon, Conceptualization, Data curation, Formal analysis, Investigation, Visualization, Methodology, Writing – original draft, Writing – review and editing; Caitlin E McDonough-Goldstein, Conceptualization, Data curation, Formal analysis, Investigation, Visualization, Writing – review and editing; Snigdha Misra, Data curation, Formal analysis, Visualization, Methodology, Writing – review and editing; Geoffrey D Findlay, Conceptualization, Data curation, Formal analysis, Investigation, Visualization, Methodology, Writing – review and editing; Andrew G Clark, Mariana Federica Wolfner, Conceptualization, Resources, Formal analysis, Supervision, Funding acquisition, Writing – original draft, Project administration, Writing – review and editing

## Author ORCIDs

Nora C Brown http://orcid.org/0000-0001-8567-1273
Benjamin Gordon https://orcid.org/0000-0002-3856-0500
Geoffrey D Findlay http://orcid.org/0000-0001-8052-2017
Andrew G Clark http://orcid.org/0000-0001-7159-8511
Mariana Federica Wolfner https://orcid.org/0000-0003-2701-9505

## Decision letter and Author response

Decision letter https://doi.org/10.7554/eLife.86409.sa1
Author response https://doi.org/10.7554/eLife.86409.sa2

# Additional files

## Supplementary files

• Supplementary file 1. gRNA sequences from flyCRISPR's Optimal Target Finder tool for each *Obp* gene.

• Supplementary file 2. Primer sequences for cloning gRNAs from *Supplementary file 1* into pAC-U63-tgRNA-Rev using pMGC as a PCR template (from *Poe et al., 2019*).

• Supplementary file 3. Primer sequences used in this study.

• Supplementary file 4. CRISPR mutant allele summary for each *Obp* gene.

• Supplementary file 5. Proportion of CS females mated to CRISPR mutant males with morphologically normal mating plugs assessed immediately after the end of mating.

• MDAR checklist

## Data availability

All data generated or analyzed for this study are included in the manuscript, supporting files, or are available on Github. Source data files have been provided for Figure 1C, Figure 2B, Figure 4C, Figure 1-figure supplement 2B, Figure 1-figure supplement 3A & B, Figure 4-figure supplement 2, and Figure 6-figure supplement 1. All mating data, R code to analyze mating data, RNAseq data across species, and tree files/alignments for use in PAML are available on Github (copy archived at *Brown et al., 2023*).

The following previously published datasets were used:

| Author(s) | Year | Dataset title | Dataset URL | Database and Identifier |
|---|---|---|---|---|
| Yang H, Jaime M, Polihronakis M, Kanegawa K, Markow T, Kaneshiro K, Oliver B | 2018 | RNA-seq of sexed adult tissues/body parts from eight *Drosophila* species | https://www.ncbi.nlm.nih.gov/geo/query/acc.cgi?acc=GSE99574 | NCBI Gene Expression Omnibus, GSE99574 |
| De Waegeneer M, Janssens J, Li H, Aerts S | 2021 | The Fly Cell Atlas: single-cell transcriptomes of the entire adult Drosophila - 10x | https://www.ebi.ac.uk/biostudies/arrayexpress/studies/E-MTAB-10519 | ArrayExpress, E-MTAB-10519 |

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

# Appendix 1

## Supplemental methods

To build our gRNA-expressing vectors, we used pAC-U63-tgRNA-Rev, a plasmid that expresses multiplexed gRNAs separated by rice Gly tRNA sequences, as well as the (*F*+E) gRNA scaffold, under the control of the *Drosophila* U6:3 promoter (**Poe et al., 2019**). We designed Gibson assembly primers containing our gRNA sequences according to **Poe et al., 2019**, **Supplementary file 2**. We used these primers to generate PCR products using the pMGC template vector and purified products of the correct size using a gel extraction kit (**Poe et al., 2019**, Zymo). The empty pAC-U63-tgRNA-Rev plasmid was digested using *Sap*I, and the digested vector and purified PCR products were assembled using the HiFi assembly kit (NEB, NEBuilder). The pAC-U63-tgRNA-Rev and pMGC plasmids were generous gifts from Chun Han at Cornell University.

## Appendix 1—key resources table

| Reagent type (species) or resource | Designation | Source or reference | Identifiers | Additional information |
|---|---|---|---|---|
| Gene (*Drosophila melanogaster*) | Obp56g | FlyBase | FLYB: FBgn0034474 | |
| Gene (*D. melanogaster*) | Obp56i | FlyBase | FLYB: FBgn0043532 | |
| Gene (*D. melanogaster*) | Obp56f | FlyBase | FLYB: FBgn0043533 | |
| Gene (*D. melanogaster*) | Obp56e | FlyBase | FLYB: FBgn0034471 | |
| Gene (*D. melanogaster*) | Obp22a | FlyBase | FLYB: FBgn0043539 | |
| Gene (*D. melanogaster*) | Obp51a | FlyBase | FLYB: FBgn0043530 | |
| Gene (*D. melanogaster*) | Obp8a | FlyBase | FLYB: FBgn0030103 | |
| Genetic reagent (*D. melanogaster*) | Obp56g1 | Bloomington *Drosophila* Stock Center | BDSC:55079; FBst0055079; RRID:BDSC_55079 | FlyBase genotype: w*; TI{GAL4}Obp56g1 |
| Genetic reagent (*D. melanogaster*) | UAS-CD4-tdGFP | Bloomington *Drosophila* Stock Center | BDSC:35836; FBst0035836; RRID:BDSC_35836 | FlyBase genotype: w1118; PBac{UAS-CD4-tdGFP}VK00033 |
| Genetic reagent (*D. melanogaster*) | Lhm; PBac{Ubnls-eGFP, ProtB-eGFP}(3) | Gift from John Belote and Scott Pitnik | | |
| Genetic reagent (*D. melanogaster*) | w;Df(2 R)/CyO | Bloomington *Drosophila* Stock Center | BDSC:25678; FBst0025678; RRID:BDSC_25678 | FlyBase genotype: w1118; Df(2 R) BSC594/CyO |
| Genetic reagent (*D. melanogaster*) | w;CyO/Bl;TM2/TM6B | Bloomington *Drosophila* Stock Center | BDSC:3704 (formerly, stock no longer available) | Genotype: w[1118]/Dp(1;Y)y[+]; CyO/Bl[1]; TM2/TM6B, Tb[1] |
| Genetic reagent (*D. melanogaster*) | nos-Cas9attP2 | BestGene/Shu Kondo & Ryu Ueda | | Genotype: y1 w1118; attP2{nos-Cas9}/TM6C,Sb Tb |
| Genetic reagent (*D. melanogaster*) | w;vasa-Cas9 | Bloomington *Drosophila* Stock Center | BDSC:51324; FBst0051324; RRID:BDSC_51324 | FlyBase genotype: w1118; PBac{vas-Cas9}VK00027 |
| Genetic reagent (*D. melanogaster*) | Obp56gRNAi | Vienna *Drosophila* Resource Center | VDRC:23206 (GD); FBst0454878 | FlyBase genotype: w1118; P{GD13268} v23206/TM3 |
| Genetic reagent (*D. melanogaster*) | CrebA-GAL4 | Bloomington *Drosophila* Stock Center | BDSC: 49409; FBst0049409; RRID:BDSC_49409 | FlyBase genotype: w1118; P{GMR64E07-GAL4}attP2 |
| Genetic reagent (*D. melanogaster*) | Tubulin-GAL4 | **Findlay et al., 2014**; 10.1371/journal.pgen.1004108 | | |
| Genetic reagent (*D. melanogaster*) | FM7c | Gift from Susan Younger | | Genotype: C(1)DX, y[1] w[1] f[1]/FM7c, Kr-GAL4[DC1], UAS-GFP[DC5], sn[+];;; |

*Appendix 1 Continued on next page*

*Appendix 1 Continued*

| Reagent type (species) or resource | Designation | Source or reference | Identifiers | Additional information |
|---|---|---|---|---|
| Genetic reagent (*D. melanogaster*) | Phi-C31 integrase attP9A | Bloomington *Drosophila* Stock Center / Rainbow Transgenics | BDSC: 35569; FBst0035569; RRID:BDSC_35569 | FlyBase genotype: y1 w* P{nanos-phiC31\int.NLS}X; PBac{y+-attP-9A} VK00027 |
| Genetic reagent (*Drosophila ananassae*) | Wildtype (Cebu, Philippines) | National *Drosophila* Species Stock Center | SKU: 14024–0371.37 | |
| Genetic reagent (*Drosophila pseudoobscura*) | Genome line | National *Drosophila* Species Stock Center | SKU: 14011–0121.94 | |
| Genetic reagent (*Drosophila mojavensis*) | Wildtype (Chocolate Mountains) | National *Drosophila* Species Stock Center | SKU: 15081–1352.00 | |
| Genetic reagent (*Drosophila virilis*) | Genome line | National *Drosophila* Species Stock Center | SKU: 15010–1051.87 | |
| Software, algorithm | CRISPR Optimal Target Finder (flyCRISPR) | *Gratz et al., 2014*; http://doi.org/10.1534/genetics.113.160713 | | |
| Genetic reagent (*D. melanogaster*) | w;;gRNA(Obp8a, ebony) | This paper | | Transgenic stock carrying gRNAs targeting Obp8a and ebony |
| Genetic reagent (*D. melanogaster*) | w;;gRNA(Obp56e, ebony) | This paper | | Transgenic stock carrying gRNAs targeting Obp56e and ebony |
| Genetic reagent (*D. melanogaster*) | w;;gRNA(Obp56f, ebony) | This paper | | Transgenic stock carrying gRNAs targeting Obp56f and ebony |
| Genetic reagent (*D. melanogaster*) | w;;gRNA(Obp56i, ebony) | This paper | | Transgenic stock carrying gRNAs targeting Obp56i and ebony |
| Genetic reagent (*D. melanogaster*) | w;;gRNA(Obp22a, ebony) | This paper | | Transgenic stock carrying gRNAs targeting Obp22a and ebony |
| Genetic reagent (*D. melanogaster*) | w;;gRNA(Obp51a, ebony) | This paper | | Transgenic stock carrying gRNAs targeting Obp51a and ebony |
| Genetic reagent (*D. melanogaster*) | Obp8aΔ390 | This paper | | CRISPR deletion of 390 bp in exon 2 of Obp8a |
| Genetic reagent (*D. melanogaster*) | Obp22aΔ257 | This paper | | CRISPR deletion of 257 bp in exon 2 of Obp22a |
| Genetic reagent (*D. melanogaster*) | Obp51aΔ16 | This paper | | CRISPR deletion of 16 bp in exon 1 of Obp51a |
| Genetic reagent (*D. melanogaster*) | Obp56eΔ239 | This paper | | CRISPR deletion of 239 bp in exon 2 of Obp56e |
| Genetic reagent (*D. melanogaster*) | Obp56fΔ226 | This paper | | CRISPR deletion of 226 bp in exon 2 of Obp56f |
| Genetic reagent (*D. melanogaster*) | Obp56iΔ359 | This paper | | CRISPR deletion of 359 bp in exon 2 of Obp56i |
| Antibody | Anti-SP (rabbit polyclonal) | Wolfner lab | | 1:1000 |
| Antibody | Anti-CG1656 (rabbit polyclonal) | Wolfner lab | | 1:500 |
| Antibody | Anti-CG1652 (rabbit polyclonal) | Wolfner lab | | 1:250 |
| Antibody | Anti-Acp36DE (rabbit polyclonal) | Wolfner lab | | 1:12,000 |
| Antibody | Anti-Acp26Aa (rabbit polyclonal) | Wolfner lab | | 1:5000 |

*Appendix 1 Continued on next page*

*Appendix 1 Continued*

| Reagent type (species) or resource | Designation | Source or reference | Identifiers | Additional information |
|---|---|---|---|---|
| Antibody | Anti-CG9997 (rabbit polyclonal) | Wolfner lab | | 1:750 |
| Antibody | Anti-Antr (rabbit polyclonal) | Wolfner lab | | 1:750 |
| Antibody | Anti-CG17575 (rabbit polyclonal) | Wolfner lab | | 1:500 |
| Antibody | Anti-tubulin (mouse monoclonal) | Sigma-Aldrich | T5168 | 1:4000 |
| Software, algorithm | RAxML-NG | *Kozlov et al., 2019*; http://doi.org/10.1093/bioinformatics/btz305 | | |
| Software, algorithm | MEGA-11 | *Tamura et al., 2021*; http://doi.org/10.1093/molbev/msab120 | | |
| Software, algorithm | PAML (v4.9) | *Yang, 2007*; https://doi.org/10.1093/molbev/msm088 | | |
| Software, algorithm | R (4.2.1) | https://www.R-project.org/ | | |

