## [Editor Report]

This important study describes an atypical role of the odorant binding protein Obp56g in mating plug formation in *Drosophila melanogaster* suggesting that Obps may play roles in reproduction in addition to their originally described roles in olfaction. Mutant males lacking Obp56g fail to induce the formation of a mating plug in the female reproductive tract-leading to ejaculate loss and reduced sperm storage. The evidence supporting the claims of the authors is solid and the work will be of interest to biologists studying Obps and seminal fluid protein function and their evolution.

---

## [Decision Letter]

**Decision letter after peer review:**

Thank you for submitting your article "The seminal odorant binding protein Obp56g is required for mating plug formation and male fertility in *Drosophila melanogaster*" for consideration by *eLife*. Your article has been reviewed by 3 peer reviewers, and the evaluation has been overseen by a Reviewing Editor and Claude Desplan as the Senior Editor. The reviewers have opted to remain anonymous.

Essential revisions:

The reviewers find your work important and of high quality. They have raised a number of concerns, however, that you will find below in their detailed comments. When preparing your revision please respond to all comments in your response letter. Moreover, in addition to more statistical analysis and quantification, please provide additional experimental support for the following points. Please consider these as essential for acceptance.

1) In Figure 2, with the exception of Opb8a, there are no wild-type controls shown. Females are mated to homozygous or heterozygous null males. Are there effects in heterozygous males? Please provide experimental evidence regarding a potential effect of heterozygosity compared to wild-type animals.

2) To support the conclusion that Obp56g is the most highly expressed seminal Obp in the ejaculatory bulb, please quantify the RT-PCR data shown in Figure 3—figure supplement 2. The data currently shown is not semi-quantitative.

3) Figure 4—figure supplement 1- Please provide quantitative data to more strongly support the claim that Obp56g1 mutant males have no sperm transfer defects during mating.

4) Please quantify the Western blot data presented in Figure 4—figure supplement 2 to support the conclusion that females mated to Obp56g1 null males have reduced SFPs in their bursas at 35 minutes ASM.

5) Please clarify the use of biological replicates (e.g., Figure 1 and Figure 1—figure supplement 1). Please explain why the data was not combined and an appropriate statistical model was used.

*Reviewer #1 (Recommendations for the authors):*

The authors convincingly demonstrate that Obp56g is expressed in the male reproductive tract and is critical for mating plug formation, and thus sperm retention and post-mating responses in females.

In its current form, several components of the manuscript seem unconnected: the analysis of Obp22a and Obp51a seem peripheral to the authors' main conclusions about the role of Obp56g in mating plug formation and are not well integrated with this finding, given that neither of these seem to affect PMR phenotypes (Figure 1 —figure supplement 1). It's also not clear how the structural information shown in Figure 5 —figure supplement 9 contributes to the overall findings.

The data regarding mating latency and duration (shown in Figure 2 —figure supplement 3) and sperm transfer (shown in Figure 4 —figure supplement 1) are critical to demonstrate that Opb56g mutant males do not show baseline mating deficits and should be moved from the supplement into the main set of figures.

In general, the text in the figures is too small/low resolution to be read clearly. In many figures, the genotype names are not legible.

In Figure 2, with the exception of Opb8a, there are no wild-type controls shown. Females are mated to homozygous or heterozygous null males. Is it possible that there are effects in heterozygous males? Data from +/cyo males are shown in Figure 1B, but it's not clear if this control is also matched to the data shown in Figure 2.

In Figure 6, D. simulans is mentioned in the text but is not shown in the figure.

*Reviewer #2 (Recommendations for the authors):*

At this point, the observation is very interesting and should certainly be reported in *eLife*. I only have a few discussion points to raise with the authors.

– I would like to invite the authors to speculate a bit more about how you think these seminal Obps collectively contribute to the sequential and temporal formation of the anterior and posterior mating plug.

– What role does the mating plug play in sperm movement in the female reproductive tract? Is there any evidence that mating plug formation activates reproductive tract stretch receptors that could contribute to the PMR?

– The ejaculatory bulb is also the site of synthesis of male pheromones, including cis-vaccenyl acetate (cVA), which is transferred to the female reproductive tract during mating. Is it imaginable that the mating plug supports the placement or concentration of pheromones at a suitable place in the female reproductive tract?

*Reviewer #3 (Recommendations for the authors):*

Figure 1-2. To strengthen the conclusion that Obp56g is a key regulator of fecundity and female remating, while Obp22a, Obp51a, Obp56e, Obp56f, Obp56i, and Obp8a do not play significant roles, it's important to conduct statistical analysis on combined data from at least three biological replicates. Although the study includes additional replicates, presenting the combined data on a single graph and performing appropriate statistical analysis would provide stronger evidence.

Figure 2: The authors examined the impact of Obp22a, Obp51a, Obp56e, Obp56f, Obp56i, and Obp8a mutations on female fecundity and remating and conclude that these genes have little or no effect. However, the study lacks a genetic control to compare with the mutant flies. Adding a control would help determine whether the removal of one copy of these Obp genes has any phenotypic effects.

It is important to note that Cyo/+ are not wild-type flies and this should be clarified in the manuscript. Have the authors confirmed that Cyo does not lead to any phenotypic effects?

In several figures presented in the study, the genotype of the control is not indicated (e.g. Figure 1—figure supplement 2; Figure 1—figure supplement 3 A-C).

In Figure 1—figure supplement 2, the authors investigate the effects of whole-body knockdown of Obp56g using Tubulin-GAL4 and conclude that it recapitulates the phenotypes observed in Obp56g1462 /Df(2R). To strengthen their conclusion, the authors should include GAL4/+ and RNAi/+ control in the experimental design and compare them with Tubulin>Obp56g RNAi.

Figure 2—figure supplement 3 A, C- The authors state that there are no significant differences in either mating latency or duration in any of the Obp56f, Obp56i, Obp56e, Obp51a, Obp22a mutant lines. To support this finding, a genetic control should be included in the analysis.

Figure 2—figure supplement 3-B, D. The authors should clearly indicate all statistical comparisons made between the experimental line and the genetic controls.

Figure 3—figure supplement 2. In order to support the conclusion that Obp56g is the most highly expressed seminal Obp in the ejaculatory bulb, the authors should include quantitative data obtained through RT-PCR analysis.

Figure 3—figure supplement 2 shows that Obp56g is not only expressed in the male ejaculatory bulb but also in the ejaculatory duct, testes, and accessory glands. Therefore, it is possible that the promoter-trap Obp56g-GAL4 does not completely reproduce the expression pattern of Obp56g in all tissues (Figure 3). This should be acknowledged in the text.

532 …'In order to test this, we crossed a ProtamineB-eGFP transgene […] into the Obp56g1 mutant line, mated null and control males to females'. What do the authors mean by 'mated null'? I can't find the corresponding data in Figures 4A and 4B.

Figure 4A-C. The authors should provide additional quantitative data to support their claim that females mated to Obp56g1 null males have defects in mating plug formation. While quantifying the proportion of females with mating plugs or sperm masses is a useful metric, it may not be sufficient to detect all potential effects. In addition, it is not clear how mating plug formation -visualized by autofluorescence- is differentiated from sperm storage -visualized using ProtB-eGFP-.

Figure 4—figure supplement 1- To support the claim that Obp56g1 mutant males have no sperm transfer defects during mating, quantitative data should be provided instead of just images of females mated to control males or Obp56g1;ProtB-eGFP mutants. It would be also informative to investigate whether the mutation affects sperm production in the testis.

Figure 1—figure supplement 3 A-C-To strengthen the conclusion that ejaculatory duct/bulb expression of Obp56g is required for mating plug formation, sperm storage, and the post-mating response, data from three independent experiments should be combined and analyzed with an appropriate statistical test. Also, the authors should provide quantitative data on mating plug formation in Figure 1—figure supplement 3B.

Figure 4—figure supplement 2. Quantification of Western blot data from three independent experiments is needed to support the conclusion that females mated to Obp56g1 null males have reduced amount of SFPs in their bursas 35 minutes ASM.

---

## [Author Response]

Essential revisions:The reviewers find your work important and of high quality. They have raised a number of concerns, however, that you will find below in their detailed comments. When preparing your revision please respond to all comments in your response letter. Moreover, in addition to more statistical analysis and quantification, please provide additional experimental support for the following points. Please consider these as essential for acceptance.1) In Figure 2, with the exception of Opb8a, there are no wild-type controls shown. Females are mated to homozygous or heterozygous null males. Are there effects in heterozygous males? Please provide experimental evidence regarding a potential effect of heterozygosity compared to wild-type animals.

The design of our mating experiments was informed by our lab’s previous work on SFPs (ovulin, SP, and Acp36DE) in which the experimental male genotype is typically homozygous null, and the control is typically heterozygous—SFPs are extremely abundant even in single copy, thus we expected the Obp mutant/CyO heterozygotes to act similarly as controls. However, we do acknowledge that our initial data did not test the full complement of +/+, +/-, and -/- combinations to address possible heterozygous effects. In order to test this, we screened through our initial CRISPR F1 line collection and isolated a line that experienced the same crossing scheme as our mutants, but did not have an editing event in the intended gene. We used this line as a +/+ control, and generated +/- mutants to compare with -/- mutants in new mating assays. Using this approach, we found no statistically significant difference in egg laying or remating rates between the three genotypes for any gene, supporting our original conclusions that loss of these genes does not impact the PMR. We have kept our original Figure 2 but have added a supplemental figure (Figure 2 figure supplement 2) and the following text to the methods and Results section to reflect these new experiments:

Methods: “We additionally isolated an unedited sibling line and crossed each Obp mutant line to compare homozygous wildtype, heterozygous mutant, and homozygous mutant males without the balancer chromosome.”

Results: “We tested whether the autosomal CRISPR mutant males showed any effect when heterozygous by testing PMR phenotypes of wildtype (+/+), heterozygous mutant (+/-) and homozygous mutant (-/-) males, and found no statistically significant impact on egg laying or remating rate for *Obp22a*, *Obp51a*, *Obp56e*, *Obp56f*, or *Obp56i* (Figure 2—figure supplement 2).”

2) To support the conclusion that Obp56g is the most highly expressed seminal Obp in the ejaculatory bulb, please quantify the RT-PCR data shown in Figure 3—figure supplement 2. The data currently shown is not semi-quantitative.

Upon revision, we determined that the single nucleus RNAseq data presented in the same figure is a better and more thorough quantification of reproductive tract expression patterns than the original RT-PCR data, given the larger sample size in the Fly Cell Atlas and straightforwardly quantitative nature of RNAseq. We therefore replaced Figure 3—figure supplement 2D with an analysis of gene expression within the FCA cluster annotated as “ejaculatory bulb” and compared expression of Obp56g vs. other seminal Obps, and found Obp56g has significantly higher expression than the other 6 genes (thus our conclusions are unchanged). We have additionally drawn attention to the main SFP-producing tissues (accessory gland main and secondary cells, ejaculatory duct, and ejaculatory bulb) in Figure 3 Figure Supp 2C with red boxes, which support our in-text conclusions about the ejaculatory duct/accessory gland expression patterns of the other seminal Obps. We have updated the results with the following text to reflect these changes:

“To determine expression patterns for the other seminal Obps, we analyzed previously published single nucleus RNAseq data of the male reproductive tract tissues from the Fly Cell Atlas (Li et al., 2022). Using this approach, we confirmed that *Obp56g* is highly expressed in the ejaculatory bulb, though we also observed expression in the ejaculatory duct and male accessory glands (Figure 3—figure supplement 2B-D), suggesting expression in tissues beyond what we observed in our GFP reporter experiment (Figure 3B). For the other six Obp genes, we observed expression primarily in the accessory gland (*Obp22a, Obp56e, Obp56i, Obp8a, Obp56f*) or ejaculatory duct (*Obp51a*) (Figure 3—figure supplement 2B, C).”

3) Figure 4—figure supplement 1- Please provide quantitative data to more strongly support the claim that Obp56g1 mutant males have no sperm transfer defects during mating.

We have performed this experiment (using the 12-minute ASM time point as in Figure 4 figure supplement 1) and found no statistically significant difference in the number of sperm transferred to the female bursa between *Obp56g^1^;ProtBGFP* and *Obp56g^1^/CyO;ProtBGFP* males, which more strongly supports our conclusions in Figure 4 Figure supp 1. This sperm count data has been incorporated into Figure 4D.

4) Please quantify the Western blot data presented in Figure 4—figure supplement 2 to support the conclusion that females mated to Obp56g1 null males have reduced SFPs in their bursas at 35 minutes ASM.

We have quantified these data, normalizing the intensity of each measured band to tubulin from the same sample (within a blot, across 3 replicate blots) and performed statistical analysis. We found overall lower band intensities for several SFPs in females mated to *Obp56g^1^* males, with a statistically significant decrease in levels of Acp36DE. We have changed the following text in the Results section to reflect this new analysis:

“Rather, we observed a lower signal intensity relative to controls in the bursa of females mated to *Obp56g^1^/Df(2R)* males for Acp36DE (and its cleavage products) at 35 minutes ASM, consistent with a defect in ejaculate retention in the mutant condition (Figure 4—figure supplement 2A, lanes 4 and 5, figure supplement 2B).”

5) Please clarify the use of biological replicates (e.g., Figure 1 and Figure 1—figure supplement 1). Please explain why the data was not combined and an appropriate statistical model was used.

We considered a biological replicate to be an independent trial of the experiment (using 15-30 individual males per biological replicate). We initially separated the mating data in the deficiency experiment due to slight differences in overall fertility (observed across all genotypes, the cause of which is unknown but is potentially environmental, as some experiments were done pre- and post-COVID lab shutdown) in one replicate, though we agree a more robust statistical analysis of all combined data would be more rigorous and strengthen the findings. We have therefore combined data from all biological replicates and analyzed the effect of male genotype on a given phenotype while controlling for replicate (using replicate as a random effect in our mixed effect linear models. The relative relationship among the four genotypes was consistent across all replicates). Doing so resulted in no differences in our conclusions, though the analysis is now stronger, and we thank the reviewers for the suggestion. We have replaced Figure 1 figure supplement 1 and removed table S5 as that data are now incorporated into Figures 1 and 2.

Reviewer #1 (Recommendations for the authors):The authors convincingly demonstrate that Obp56g is expressed in the male reproductive tract and is critical for mating plug formation, and thus sperm retention and post-mating responses in females.In its current form, several components of the manuscript seem unconnected: the analysis of Obp22a and Obp51a seem peripheral to the authors' main conclusions about the role of Obp56g in mating plug formation and are not well integrated with this finding, given that neither of these seem to affect PMR phenotypes (Figure 1 —figure supplement 1). It's also not clear how the structural information shown in Figure 5 —figure supplement 9 contributes to the overall findings.

We did not find evidence to suggest a role for either *Obp22a* or *Obp51a* in the PMR, but we did not measure all aspects of reproduction or sperm competition in which they might have a role (and in which their rapid evolution might suggest a function). We have added the following text to provide additional context to the results and discussion to better support our interpretation of positive selection on *Obp22a* and *Obp51a* which we hope connects our findings more effectively:

Discussion: “Alternatively, it is possible that the genes for which we detected no PMR phenotype are involved in another aspect of reproduction or mating. Given that we detected positive selection acting on *Obp22a* and *Obp51a*, it would be informative to test whether these genes might be involved in mediating outcomes of sperm competition, as has been observed for other SFPs that show signatures of selection (Avila et al., 2010; Patlar and Civetta 2022; Wong et al., 2008). Measuring short-term remating rates (0-5 hours, before the long-term SP response becomes active) would also be informative and might be consistent with a male-derived pheromonal function for these genes (Bretman et al., 2010; Laturney and Billeter, 2016).”

Additionally, we have added the following text to the results to provide additional context for our investigation of the structure in Figure 5 figure supplement 5:

“Previous work has found that pheromones derived from the male reproductive tract and transferred during mating rapidly turn over across the *Drosophila* clade, with many of these pheromones functioning as anti-aphrodisiacs in mated females (Khallaf et al., 2021). Given this observation, we were curious if we could infer specific sites under selection (and the 3D location of these sites within the protein) to determine whether we observe changes in the binding pocket of the protein that might be consistent with changes in ejaculate-derived ligands across species. We therefore used model M8 to infer specific sites under selection for *Obp22a* and *Obp51a* (Figure 5B).”

The data regarding mating latency and duration (shown in Figure 2 —figure supplement 3) and sperm transfer (shown in Figure 4 —figure supplement 1) are critical to demonstrate that Opb56g mutant males do not show baseline mating deficits and should be moved from the supplement into the main set of figures.

We agree and have now moved these figures (and new data from the suggested experiment to count sperm transferred during mating) into Figure 1 D and E (latency and duration) and Figure 4D (sperm transfer) of the main text.

In general, the text in the figures is too small/low resolution to be read clearly. In many figures, the genotype names are not legible.

We apologize for this. We have increased the text size of all labels in our graphs and supplied Illustrator files for all figures, which should fix the low resolution issue.

In Figure 2, with the exception of Opb8a, there are no wild-type controls shown. Females are mated to homozygous or heterozygous null males. Is it possible that there are effects in heterozygous males? Data from +/cyo males are shown in Figure 1B, but it's not clear if this control is also matched to the data shown in Figure 2.

The design of our mating experiments was informed by our lab’s previous work on SFPs (ovulin, SP, and Acp36DE) in which the experimental male genotype is typically homozygous null, and the control is typically heterozygous—SFPs are extremely abundant even in single copy, thus we expected the Obp mutant/CyO heterozygotes to act similarly as controls. However, we do acknowledge that our initial data did not test the full complement of +/+, +/-, and -/- combinations to address possible heterozygous effects. In order to test this, we screened through our initial CRISPR F1 line collection and isolated a line that experienced the same crossing scheme as our mutants, but did not have an editing event in the intended gene. We used this line as a +/+ control, and generated +/- mutants to compare with -/- mutants in new mating assays. Using this approach, we found no statistically significant difference in egg laying or remating rates between the three genotypes for any gene, supporting our original conclusions that loss of these genes does not impact the PMR. We have kept our original Figure 2 but have added a supplemental figure (Figure 2 figure supplement 2) and the following text to the methods and Results section to reflect these new experiments:

Methods: “We additionally isolated an unedited sibling line and crossed each Obp mutant line to compare homozygous wildtype, heterozygous mutant, and homozygous mutant males without the balancer chromosome.”

Results: “We tested whether the autosomal CRISPR mutant males showed any effect when heterozygous by testing PMR phenotypes of wildtype (+/+), heterozygous mutant (+/-) and homozygous mutant (-/-) males, and found no statistically significant impact on egg laying or remating rate for *Obp22a*, *Obp51a*, *Obp56e*, *Obp56f*, or *Obp56i* (Figure 2—figure supplement 2).”

In Figure 6, D. simulans is mentioned in the text but is not shown in the figure.

We thank the reviewer for noting this error. We have deleted the mention of *D. simulans* in the text (we do not present data in this paper on *D. simulans* expression).

Reviewer #2 (Recommendations for the authors):At this point, the observation is very interesting and should certainly be reported in eLife. I only have a few discussion points to raise with the authors.– I would like to invite the authors to speculate a bit more about how you think these seminal Obps collectively contribute to the sequential and temporal formation of the anterior and posterior mating plug.

We thank this reviewer for the opportunity to expand our speculation, and have added the following to our discussion to reflect on this interesting point (though we note that lack of data on this process in *Drosophila* in general makes it difficult):

“Our finding that *Obp56g^1^* mutant males lack a mating plug at 12 minutes ASM suggests that this protein (and potentially its ligand, if it has one) likely functions relatively early and is required for full plug formation while the flies are still copulating.”

– What role does the mating plug play in sperm movement in the female reproductive tract? Is there any evidence that mating plug formation activates reproductive tract stretch receptors that could contribute to the PMR?

Previous work in mammals shows that prostaglandins present in the seminal fluid induce changes in musculature contractions of the female tract, which are hypothesized to help sperm movement (Wånggren et al., 2008). In flies, previous work from our lab has shown that receipt of SFPs in total (but not sperm) is sufficient to induce muscular contractions that change the conformation of the bursa to facilitate sperm storage (Adams and Wolfner, 2007). Additionally, previous work from our lab (Avila and Wolfner, 2017, 2009) has shown that two mating plug proteins, one of which is found in the anterior mating plug, the other in the posterior mating plug (Acp36DE and PEB-me, respectively) are also required for this conformational change, indicating that indeed, the mating plug does help with sperm movement into storage (though it does so by affecting the female tract, not the sperm itself). These two regions of the mating plug are also thought to “constrain” the sperm in the upper and lower regions of the bursa for maximal sperm storage and minimal sperm loss, respectively (Avila et al., 2015; Bertram et al., 1996). In the bursa, it is unknown what potential mechanosensory pathways could be acting to induce the muscular contractions. However, in the oviduct, it is known that the SFP ovulin relaxes oviduct musculature via octopaminergic signaling in neurons to facilitate oviposition (Rubinstein and Wolfner, 2013). A similar mechanism may be at play in the bursa.

– The ejaculatory bulb is also the site of synthesis of male pheromones, including cis-vaccenyl acetate (cVA), which is transferred to the female reproductive tract during mating. Is it imaginable that the mating plug supports the placement or concentration of pheromones at a suitable place in the female reproductive tract?

Data from *Drosophila* suggests the presence of cVA within the female bursa (as well as its ejection out) is associated with the mating plug (Laturney and Billeter, 2016), though the localization within the bursa (i.e. is it within the sperm mass, the anterior mating plug, or the posterior mating plug) is unknown. Thus, it is unclear if the mating plug is required for the concentration of cVA (and conversely, if cVA needs to be in a specific place within the bursa to elicit its pheromonal effects). Previous data also shows males sense cVA in recently mated females through olfactory pathways (Ha and Smith, 2006), so it is a compelling hypothesis that the mating plug might be important for concentrating anti-aphrodisiac pheromones in a region that would be “sampled” by courting males (but this is speculation).

Reviewer #3 (Recommendations for the authors):Figure 1-2. To strengthen the conclusion that Obp56g is a key regulator of fecundity and female remating, while Obp22a, Obp51a, Obp56e, Obp56f, Obp56i, and Obp8a do not play significant roles, it's important to conduct statistical analysis on combined data from at least three biological replicates. Although the study includes additional replicates, presenting the combined data on a single graph and performing appropriate statistical analysis would provide stronger evidence.

We agree with this reviewer and thank them for the suggestion. We have combined all replicates for the respective genotypes (including replicate as a random effect in our linear models) and conducted statistical analysis, and our results are unchanged (though the analysis is now stronger).

Figure 2: The authors examined the impact of Obp22a, Obp51a, Obp56e, Obp56f, Obp56i, and Obp8a mutations on female fecundity and remating and conclude that these genes have little or no effect. However, the study lacks a genetic control to compare with the mutant flies. Adding a control would help determine whether the removal of one copy of these Obp genes has any phenotypic effects.

The design of our initial mating assays was informed by our lab’s previous work on SFP function, in which homozygous mutant males are compared with heterozygous control males (which are often over balancer chromosomes, such as SP or Ovulin). We disagree with the argument that the data presented in Figure 2 lacks a control (as the balancer males went through the same crossing scheme and thus are a reasonable genetic background match), though we agree a better control would compare WT and mutant chromosomes as discussed above. This new data is in Figure 1 figure supplement 1 and confirms our initial findings that these genes have little or no effect on PMR phenotypes.

It is important to note that Cyo/+ are not wild-type flies and this should be clarified in the manuscript. Have the authors confirmed that Cyo does not lead to any phenotypic effects?

We agree and have removed the mention of wildtype associated with *+/CyO*, instead noting more precisely that these males have 2 copies of *Obp56g*.

We have not explicitly tested PMR phenotypes of *CyO/+* vs. *+/+* genotypes, but we did not observe any statistically significant differences in the PMR (in either egg laying or remating rates) of females mated to *Df2R/+* vs. either *CyO/+* or *Obp56g^1^/CyO*, suggesting that the presence of CyO does not significantly impact the male’s ability to induce normal PMR phenotypes.

In several figures presented in the study, the genotype of the control is not indicated (e.g. Figure 1—figure supplement 2; Figure 1—figure supplement 3 A-C).

We apologize for this oversight and have added the appropriate labels to each supplemental figure indicated.

In Figure 1—figure supplement 2, the authors investigate the effects of whole-body knockdown of Obp56g using Tubulin-GAL4 and conclude that it recapitulates the phenotypes observed in Obp56g1462 /Df(2R). To strengthen their conclusion, the authors should include GAL4/+ and RNAi/+ control in the experimental design and compare them with Tubulin>Obp56g RNAi.

We apologize for not making it clear that the genotype of the control line is RNAi/+ (we have fixed this in the figure legend and figure). We agree with the importance of testing GAL4 and RNAi transgenics in isolation, but our mutant data for *Obp56g* is a stronger/cleaner experiment, and suggests the PMR phenotypes we observe in the experimental condition are not an artefact of either transgene (thus we did not feel the need to additionally test GAL4/+).

Figure 2—figure supplement 3 A, C- The authors state that there are no significant differences in either mating latency or duration in any of the Obp56f, Obp56i, Obp56e, Obp51a, Obp22a mutant lines. To support this finding, a genetic control should be included in the analysis.

We have repeated these experiments using +/+, +/-, and -/- males and found no effect on latency or duration for *Obp56f, Obp22a*, and *Obp51a*; no effect on duration for *Obp56e*. We observed a slight increase in latency (-/- vs. +/-) and slight decrease in duration (-/- vs. +/- and -/- vs. +/+) for Obp56i, and a slight increase in latency (-/- vs. +/+) for *Obp56e*. These data are found in Figure 2—figure supplement 5 and we have added the following text to the Results section:

“Comparisons of latency and duration in (+/+), (+/-), and (-/-) CRISPR mutant males resulted in largely consistent results, with no effect on either phenotype for *Obp56f, Obp22a, Obp51a*, and no effect on mating duration for *Obp56e* (Figure 2—figure supplement 5). However, we did observe a slight increase in mating latency (-/- vs. +/-) and a slight decrease in duration (-/- vs. +/- and -/- vs. +/+) for *Obp56i*, and a slight increase in latency (-/- vs. +/+) for *Obp56e* (Figure 2—figure supplement 5).”

Figure 2—figure supplement 3-B, D. The authors should clearly indicate all statistical comparisons made between the experimental line and the genetic controls.

We apologize—the line drawn on the plot made the statistical comparisons unclear in Figure 2 Figure supp BandD (which is now moved to Figure 1 DandE). We analyzed the effect of male genotype on latency and duration in a mixed effects linear model, following up with pairwise comparisons on estimated marginal means between genotypes. None of these comparisons were significantly different, and we have removed the line on the plot which might have suggested that we did not test all combinations. We have added text to the results and figure legend to indicate no significant effect of genotype was observed.

Figure 3—figure supplement 2. In order to support the conclusion that Obp56g is the most highly expressed seminal Obp in the ejaculatory bulb, the authors should include quantitative data obtained through RT-PCR analysis.

Upon revision, we determined that the single nucleus RNAseq data presented in the same figure is a better and more thorough quantification of reproductive tract expression patterns than the original RT-PCR data, given the larger sample size in the Fly Cell Atlas and straightforwardly quantitative nature of RNAseq. We therefore replaced Figure 3—figure supplement 2D with an analysis of gene expression within the FCA cluster annotated as “ejaculatory bulb” and compared expression of Obp56g vs. other seminal Obps, and found Obp56g has significantly higher expression than the other 6 genes (thus our conclusions are unchanged). We have additionally drawn attention to the main SFP-producing tissues (accessory gland main and secondary cells, ejaculatory duct, and ejaculatory bulb) in Figure 3 Figure Supp 2C with red boxes, which support our in-text conclusions about the ejaculatory duct/accessory gland expression patterns of the other seminal Obps. We have updated the results with the following text to reflect these changes:

“To determine expression patterns for the other seminal Obps, we analyzed previously published single nucleus RNAseq data of the male reproductive tract tissues from the Fly Cell Atlas (Li et al., 2022). Using this approach, we confirmed that *Obp56g* is highly expressed in the ejaculatory bulb, though we also observed expression in the ejaculatory duct and male accessory glands (Figure 3—figure supplement 2B-D), suggesting expression in tissues beyond what we observed in our GFP reporter experiment (Figure 3B). For the other six Obp genes, we observed expression primarily in the accessory gland (Obp22a, Obp56e, Obp56i, Obp8a, Obp56f) or ejaculatory duct (Obp51a) (Figure 3—figure supplement 2B, C).”

Figure 3—figure supplement 2 shows that Obp56g is not only expressed in the male ejaculatory bulb but also in the ejaculatory duct, testes, and accessory glands. Therefore, it is possible that the promoter-trap Obp56g-GAL4 does not completely reproduce the expression pattern of Obp56g in all tissues (Figure 3). This should be acknowledged in the text.

We agree and have added the following to the results to acknowledge this point:

“Using this approach, we confirmed that *Obp56g* is highly expressed in the ejaculatory bulb, though we also observed expression in the ejaculatory duct and male accessory glands (Figure 3—figure supplement 2B-D), suggesting the promoter trap does not fully recapitulate Obp56g expression in all reproductive tract tissues (Figure 3B).”

532 …'In order to test this, we crossed a ProtamineB-eGFP transgene […] into the Obp56g1 mutant line, mated null and control males to females'. What do the authors mean by 'mated null'? I can't find the corresponding data in Figures 4A and 4B.

We apologize for this issue of clarity in the text, we have replaced this sentence with the following to clarify:

“In order to test this, we crossed a *ProtamineB*-eGFP transgene (Manier et al., 2010), which marks the heads of sperm with GFP, into the *Obp56g^1^* mutant line, and mated homozygous null (*Obp56g^1^*) or control (*Obp56g^1^/CyO*) males to females, and directly counted sperm in the female sperm storage organs at 3 hours and 4 days ASM”

Figure 4A-C. The authors should provide additional quantitative data to support their claim that females mated to Obp56g1 null males have defects in mating plug formation. While quantifying the proportion of females with mating plugs or sperm masses is a useful metric, it may not be sufficient to detect all potential effects. In addition, it is not clear how mating plug formation -visualized by autofluorescence- is differentiated from sperm storage -visualized using ProtB-eGFP-.

We agree that there may be additional effects on the mating plug that could be quantified, though the presence/absence phenotype of the mating plug is quite strong and easily apparent (and to our knowledge there is no other measurable metric that has been used before to describe the mating plug in *D. melanogaster*, aside from possibly gas chromatography/mass spectrometry approaches to quantify EB-derived male pheromones [which is beyond the scope of this paper]). We also note that due to the complete lack of the mating plug in mates of Obp56g1 null males, we cannot reliably measure its size.

To address the second point, a mating plug and a GFP-labeled sperm head are quite distinct even when visualized in the same fluorescence channel, due to large differences in their size, shape, and level of fluorescence (we estimate the size of the mating plug to be roughly ~200um wide X ~500um long, while GFP-labeled sperm heads are ~10um). Additionally, the location of each is distinct within the female reproductive tract—while the mating plug is only found in the posterior region of the bursa, sperm within the storage organs are within the anterior region.

Figure 4—figure supplement 1- To support the claim that Obp56g1 mutant males have no sperm transfer defects during mating, quantitative data should be provided instead of just images of females mated to control males or Obp56g1;ProtB-eGFP mutants. It would be also informative to investigate whether the mutation affects sperm production in the testis.

We have performed this experiment (using the 12-minute ASM time point as in Figure 4 figure supplement 1) and found no statistically significant difference in the number of sperm transferred to the female bursa between *Obp56g^1^;ProtBGFP* and *Obp56g^1^/CyO;ProtBGFP* males, which more strongly supports our conclusions in Figure 4 Figure supp 1. This sperm count data has been incorporated into Figure 4D.

We have addressed the second point of sperm production by imaging fixed, F-actin stained testes of 3-4 day old *Obp56g^1^/CyO;ProtB-GFP* and *Obp56g^1^;ProtB-GFP* males--the same genotypes as used in the sperm count experiments (Figure 4D). We observed normal testis morphology, normal individualization cones throughout the testis, and fully mature sperm in the seminal vesicles of both genotypes, suggesting there is no obvious defect in sperm production underlying the PMR defects of *Obp56g^1^* null males. These data have been added as a supplemental figure (Figure 1 figure supplement 3).

Figure 1—figure supplement 3 A-C-To strengthen the conclusion that ejaculatory duct/bulb expression of Obp56g is required for mating plug formation, sperm storage, and the post-mating response, data from three independent experiments should be combined and analyzed with an appropriate statistical test. Also, the authors should provide quantitative data on mating plug formation in Figure 1—figure supplement 3B.

We have performed these experiments and have included them in the data (all results were consistent with our previous conclusions). We were unable to provide more quantitative data on mating plug formation as suggested (see point above for the discussion).

Figure 4—figure supplement 2. Quantification of Western blot data from three independent experiments is needed to support the conclusion that females mated to Obp56g1 null males have reduced amount of SFPs in their bursas 35 minutes ASM.

We have quantified these data, normalizing the intensity of each measured band to tubulin from the same sample (within a blot, across 3 replicate blots) and performed statistical analysis. We found overall lower band intensities for several SFPs in females mated to Obp56g1 males, with a statistically significant decrease in levels of Acp36DE. We have changed the following text in the Results section to reflect this new analysis:

“Rather, we observed a lower signal intensity relative to controls in the bursa of females mated to *Obp56g^1^/Df(2R)* males for Acp36DE (and its cleavage products) at 35 minutes ASM, consistent with a defect in ejaculate retention in the mutant condition (Figure 4—figure supplement 2A, lanes 4 and 5, figure supplement 2B).”